# Characterizing and controlling CRISPR repair outcomes in nondividing human cells

Gokul N. Ramadoss [1,2], Samali J. Namaganda[1], Manasi M. Kumar [1], Jennifer R. Hamilton [3,4], Rohit Sharma [3,5], Karena G. Chow[1], Luke A. Workley[1,6], Bria L. Macklin[1], Mengyuan Sun[1], Alvin S. Ha [1,6], Jia-Cheng Liu[7], Christof Fellmann [1,8], Hannah L. Watry[1], Philip H. Dierks [1], Rudra S. Bose[2], Julianne Jin [2], Barbara S. Perez [3,4], Cindy R. Sandoval Espinoza [3,4], Madeline P. Matia [1], Serena H. Lu [1], Luke M. Judge[1,9], Brian R. Shy [1,6,10], Andre Nussenzweig[7], Britt Adamson [11,12], Niren Murthy [3,5], Jennifer A. Doudna [1,3,4,13,14,15,16], Martin Kampmann [2,17] & Bruce R. Conklin [1,3,8,18] ✉

Genome editing is poised to revolutionize treatment of genetic diseases, but poor understanding and control of DNA repair outcomes hinders its therapeutic potential. DNA repair is especially understudied in nondividing cells like neurons, limiting the efficiency and precision of genome editing in many clinically relevant tissues. Here, we address this barrier by using induced pluripotent stem cells (iPSCs) and iPSC-derived neurons to examine how postmitotic human neurons repair Cas9-induced DNA damage. CRISPR editing outcomes differ dramatically in neurons compared to genetically identical dividing cells: neurons take longer to fully resolve this damage, and upregulate non-canonical DNA repair factors in the process. Manipulating this response with chemical or genetic perturbations allows us to direct DNA repair toward desired editing outcomes in nondividing human neurons, cardiomyocytes, and primary T cells. By studying DNA repair in clinically relevant cells, we reveal unforeseen challenges and opportunities for precise therapeutic editing.

Thousands of genetic diseases could be corrected by precise genomic edits, using tools such as CRISPR-Cas9 to induce perturbations at targeted locations in the genome[1,2]. However, a fundamental roadblock is our inability to control how those perturbations are repaired[3].

CRISPR nucleases, base editors, and prime editors perturb DNA in different ways[4–7], but in each case, the editing outcome is ultimately determined by how the cellular DNA repair machinery responds to that perturbation[8–10]. Repair that restores the original sequence instead of

[1]Gladstone Institutes, San Francisco, CA, USA. [2]Institute for Neurodegenerative Diseases, University of California, San Francisco, CA, USA. [3]Innovative Genomics Institute, University of California, Berkeley, CA, USA. [4]Department of Molecular & Cell Biology, University of California, Berkeley, CA, USA. [5]Department of Bioengineering, University of California, Berkeley, CA, USA. [6]Department of Laboratory Medicine, University of California, San Francisco, CA, USA. [7]Laboratory of Genome Integrity, National Cancer Institute, NIH, Bethesda, MD, USA. [8]Department of Cellular & Molecular Pharmacology, University of California, San Francisco, CA, USA. [9]Department of Pediatrics, University of California, San Francisco, CA, USA. [10]Helen Diller Family Comprehensive Cancer Center, University of California, San Francisco, CA, USA. [11]Department of Molecular Biology, Princeton University, Princeton, NJ, USA. [12]Lewis–Sigler Institute for Integrative Genomics, Princeton University, Princeton, NJ, USA. [13]California Institute for Quantitative Biosciences, University of California, Berkeley, CA, USA. [14]Howard Hughes Medical Institute, University of California, Berkeley, CA, USA. [15]Department of Chemistry, University of California, Berkeley, CA, USA. [16]MBIB Division, Lawrence Berkeley National Laboratory, Berkeley, CA, USA. [17]Department of Biochemistry & Biophysics, University of California, San Francisco, CA, USA. [18]Department of Medicine, University of California, San Francisco, CA, USA. ✉e-mail: bconklin@gladstone.ucsf.edu

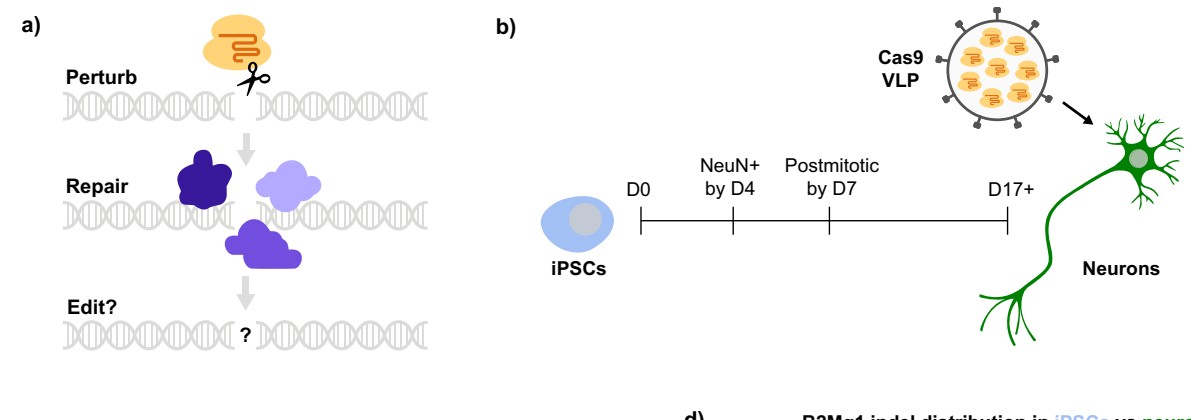

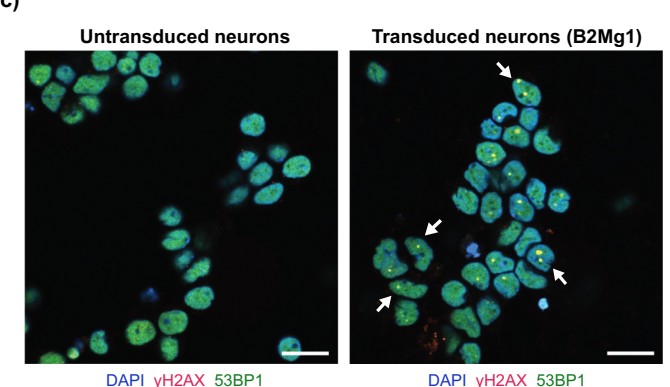

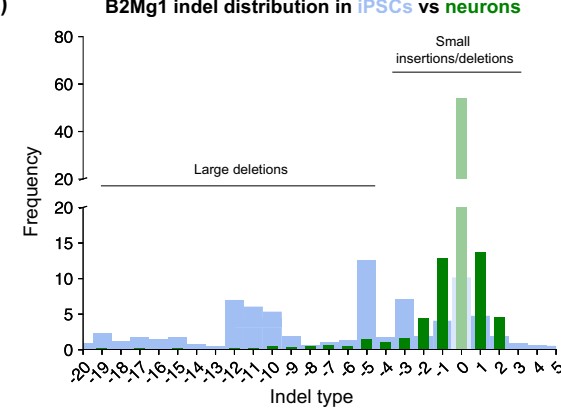

**Fig. 1 | Modeling CRISPR repair outcomes in postmitotic human neurons.**
**a** Schematic: Genome editing proteins can perturb DNA, but cellular DNA repair determines the editing outcome. **b** Timeline of differentiating iPSCs (blue) into neurons (green). After at least 2 weeks of differentiation/maturation, postmitotic neurons are treated with VLPs delivering Cas9 protein (yellow) and sgRNA (orange). **c** Cas9 VLPs induce DSBs in human iPSC-derived neurons. Representative ICC images of neurons 3 days post-transduction with B2Mg1 VLPs, and age-matched untransduced neurons. Scale bar is 20 μm. Arrows denote examples of DSB foci: yellow puncta co-labeled by γH2AX (red) and 53BP1 (green). Dose: 1 μL VLP (FMLV) per 100 μL media. Additional images shown in Supplementary Fig. 5, with similar results replicated independently in Supplementary Fig. 12. **d** Genome editing outcomes differ between iPSCs and isogenic neurons. CRISPResso2 analysis of amplicon-NGS, from cells 4 days post-transduction with B2Mg1 VLPs. Dose: 2 μL VLP (HIV) per 100 μL media. Data are averaged across 6 replicate wells per cell type transduced in parallel, and expressed as a percentage of total reads. Thick blue background bars are from iPSCs; thin green foreground bars are from neurons.

editing it is unproductive, and imprecise repair can cause harmful unintended changes[3]. To ensure that the desired edit occurs in each cell, therapeutic genome editing requires a thorough understanding and control of DNA repair.

Surprisingly little is known about DNA repair in postmitotic cells such as neurons, which cannot regenerate yet must withstand an entire lifetime's worth of DNA damage. This gap in understanding hinders research into many diseases, such as neurodegeneration and aging, and also limits our control over CRISPR editing outcomes. Many neurodegenerative diseases are caused by dominant genetic mutations, making them strong candidates for CRISPR-based gene inactivation[11–16]. Cas9-induced double-strand breaks (DSBs) can disrupt these mutant alleles and reverse disease phenotypes. However, this requires specific DSB repair outcomes that produce the proper insertion/deletion mutations (indels) capable of frameshifting and eliminating the toxic gene product[17].

Whether the DSB results in a desired indel or not is determined by the competing DSB repair pathways active in the cell (Fig. 1a and Supplementary Fig. 1). In fact, differential expression of even a single DNA repair gene can change a cell's editing outcome[8]. DSB repair pathways in nondividing cells likely differ drastically from those in the rapidly-proliferating and transformed cell lines used by most editing studies to date[18–21]. Pathways such as homology directed repair (HDR), for example, which are restricted to certain stages of the cell cycle, should be inactive in non-cycling cells[22]. Furthermore, DSB repair may be particularly unique in neurons, where some early-response genes are activated by the presence of DSBs in their own promoters[20], and DSBs have even been implicated in memory formation[23]. Therefore, the rules of CRISPR editing outcomes may differ in postmitotic neurons compared to the dividing cells that have shaped the literature thus far.

To test this, in this study, we compare how human induced pluripotent stem cells (iPSCs) and iPSC-derived neurons respond to Cas9-induced DNA damage. Compared to these isogenic dividing cells, neurons accumulate indels over a longer time period and upregulate unexpected DNA repair genes in response to Cas9 exposure. Manipulating this repair response allows us to influence the efficiency and/or precision of genome editing in postmitotic neurons and cardiomyocytes, and in nondividing primary human T cells—adding important new tools to the genome modification toolkit.

## Results

### Virus-like particles efficiently deliver Cas9 to human iPSC-derived neurons

To investigate how Cas9-induced DSBs are repaired in neurons, we first needed a platform to deliver controlled amounts of Cas9 into postmitotic human neurons. We used a well-characterized protocol[24,25] to differentiate human iPSCs into cortical-like excitatory neurons (Fig. 1b). Immunocytochemistry (ICC) confirmed the purity of these iPSC-derived neurons. Over 99% of cells were Ki67-negative by Day 7 of differentiation, and approximately 95% of cells were NeuN-positive from Day 4 onward (Supplementary Fig. 2). These observations

confirm that within one week our cells rapidly become postmitotic, and uniformly express key neuronal markers.

While iPSCs and other dividing cells are amenable to electroporation and chemical transfection, transient Cas9 delivery to neurons remains challenging. Recently, virus-like particles (VLPs) inspired by Friend murine leukemia virus (FMLV), human immunodeficiency virus (HIV), and others have been used to successfully deliver CRISPR enzymes to many mouse tissues, including mouse brain[26–29]. Unlike viruses, which deliver genomic material into cells, VLPs are engineered to deliver protein cargo such as Cas9. Viruses pseudotyped with the glycoprotein VSVG are known to transduce LDLR-expressing cells, including neurons[30], and co-pseudotyping particles with the envelope protein BaEVRless (BRL) has been shown to improve transduction in multiple human cell types[31]. Therefore, we reasoned that VLPs pseudotyped with VSVG and/or BRL could efficiently transduce human neurons.

We produced VLPs containing Cas9 ribonucleoprotein (RNP) to induce DSBs, with or without an mNeonGreen transgene to track transduction. By flow cytometry, we found that multiple types of VLPs effectively delivered cargo to our neurons, with up to 97% efficiency (Supplementary Fig. 3). Additionally, modulating the VLP's pseudotype or the Cas9's nuclear localization sequence both greatly impacted delivery efficiency (Supplementary Fig. 4). For subsequent experiments, we proceeded with two particles interchangeably: VSVG-pseudotyped HIV VLPs (also known as enveloped delivery vehicles[27]), or VSVG/BRL-co-pseudotyped FMLV VLPs. Furthermore, ICC confirmed that Cas9-VLPs successfully induced DSBs in our neurons, co-labeled by markers gamma-H2AX (γH2AX) and 53BP1 (Fig. 1c and Supplementary Fig. 5). This platform to acutely perturb DNA in human neurons enables the study of DNA repair in clinically relevant postmitotic cells.

## CRISPR repair outcomes differ in neurons compared to dividing cells

To examine how neurons repair DSBs, we used VLPs to deliver equal doses of Cas9 RNP into human iPSC-derived neurons and genetically identical iPSCs. We selected a single-guide RNA (sgRNA), B2Mg1, that yields a variety of indel types in iPSCs, suggesting it is compatible with multiple DSB repair pathways. End resection-dependent DSB repair pathways, such as microhomology-mediated end joining (MMEJ), are typically restricted to certain stages of the cell cycle (S/G2/M), while nonhomologous end joining (NHEJ) is not[22,32,33]. Since postmitotic cells have exited the cell cycle, they are predicted to predominantly utilize NHEJ when repairing DSBs.

Indeed, while B2Mg1-edited iPSCs displayed a broad range of indels, neurons exhibited a much narrower distribution of outcomes (Fig. 1d). In iPSCs, the most prevalent indel outcomes were larger deletions typically associated with MMEJ, as expected for dividing cells[33]. In neurons, the most prevalent outcomes were those usually attributed to NHEJ: small indels associated with NHEJ processing, and unedited outcomes caused by either indel-free classical NHEJ (cNHEJ) or lack of Cas9 cutting[34,35]. This was true for several different sgRNAs tested. Even though each sgRNA had a different intrinsic distribution of available indel types, in each case, the MMEJ-like larger deletions were predominant in iPSCs, and the NHEJ-like smaller indels were predominant in neurons. Therefore, for every sgRNA we tested, the ratio of insertions to deletions was significantly higher in neurons than iPSCs (Supplementary Fig. 6). These results demonstrate that postmitotic neurons employ different DSB repair pathways than dividing cells, yielding different CRISPR editing outcomes.

Unresolved DSBs can be lethal to cycling cells, as DNA damage checkpoints trigger cell cycle arrest and/or apoptosis[36,37]. Therefore, for dividing cells, resolving a DSB mutagenically can be less harmful than leaving it unrepaired. For example, mitotic cells often utilize extremely indel-prone MMEJ repair to avoid progressing through M

phase with unresolved DSBs[32,33]. This is consistent with our observed editing outcomes in iPSCs. On the other hand, postmitotic cells do not face replication checkpoints, and thus might not be subjected to the same pressures. Therefore, we hypothesized that DSBs could be resolved over a longer time scale in postmitotic cells.

## Cas9-induced indels accumulate slowly in neurons

In dividing cells, the repair half-life of Cas9-induced DSBs is reportedly between 1 and 10 h; even in the slowest-repaired cut sites, the fraction of unresolved DSBs peaks within just over 1 day[38]. DSB repair in our iPSCs matched this expected timing, with indels plateauing within a few days. In contrast, indels in neurons continued to increase for up to 2 weeks post-transduction (Fig. 2a).

This extended time course of editing was replicated by both types of VLPs (Supplementary Fig. 7). We tested multiple sgRNAs, including disease-relevant targets. Surprisingly, for every sgRNA, neuron indels continued to increase for at least 16 days post-delivery of transient Cas9 RNP (Fig. 2b and Supplementary Fig. 8). We also observed a similar weeks-long timeline of indel accumulation in postmitotic iPSC-derived cardiomyocytes (Supplementary Fig. 9a), so this prolonged indel accumulation might also apply to other clinically relevant nondividing cells besides neurons.

We found no evidence that this prolonged indel accumulation in neurons was influenced by proliferating cells (Supplementary Fig. 2), or by residual VLP in the media (Supplementary Fig. 9b). Furthermore, using the same delivery particle but engaging a different DNA repair pathway than DSBs, VLP-mediated base editing in neurons was comparably efficient to iPSCs—and sometimes even more efficient— even within only three days post transduction (Supplementary Fig. 9c). Therefore, the slower accumulation of indels cannot be attributed solely to a "delivery deficit" in neurons.

However, the kinetics of VLP entry and trafficking could still play a role in the prolonged time course of editing. To test this, we used another model of genetically identical dividing and nondividing cells: primary human T cells in the activated vs resting state. While resting T cells are not amenable to VLP delivery, both resting and activated T cells are amenable to electroporation, unlike neurons—enabling Cas9 RNP delivery without encapsulation in a delivery particle. Therefore, we electroporated Cas9 RNP directly into activated or resting primary T cells from multiple human donors, circumventing VLP delivery kinetics entirely. In this model, while activated vs resting T cells reproduced the observed differences in indel types between dividing and nondividing cells, there was *not* a dramatic difference in the timing of indels (Supplementary Figs. 10 and 11). This suggests that a component of the prolonged indel accumulation observed in postmitotic neurons and cardiomyocytes could be related to delivery kinetics, and/or dependent on cell type.

This week-long timeline of editing in postmitotic cells could have major clinical implications. Gene inactivation therapies in nondividing tissues might take longer than anticipated to be effective, and both on-target and off-target editing may accumulate over longer intervals. Additionally, persistent DSBs in neurons have been associated with genomic instability and even neurodegeneration[39–41], so characterizing the duration of Cas9-induced damage and repair is critical.

## DSB repair is detectable in neurons for more than 1 week post-Cas9 delivery

To assess the duration of this damage in neurons, we measured multiple signals of DSB repair over time after delivering transient Cas9 RNP via VLPs. DSB foci (γH2AX/53BP1) were strongly detectable by ICC as early as 1 day post-transduction, confirming efficient delivery and rapid induction of DSBs in neurons. Interestingly, DSB foci remained detectable in neurons for at least 7 days post-transduction (Fig. 2c–e). Persistent DSB repair signal was observed for sgRNAs targeting both lowly-transcribed (*B2M*) and highly-transcribed (*NEFL*) genes

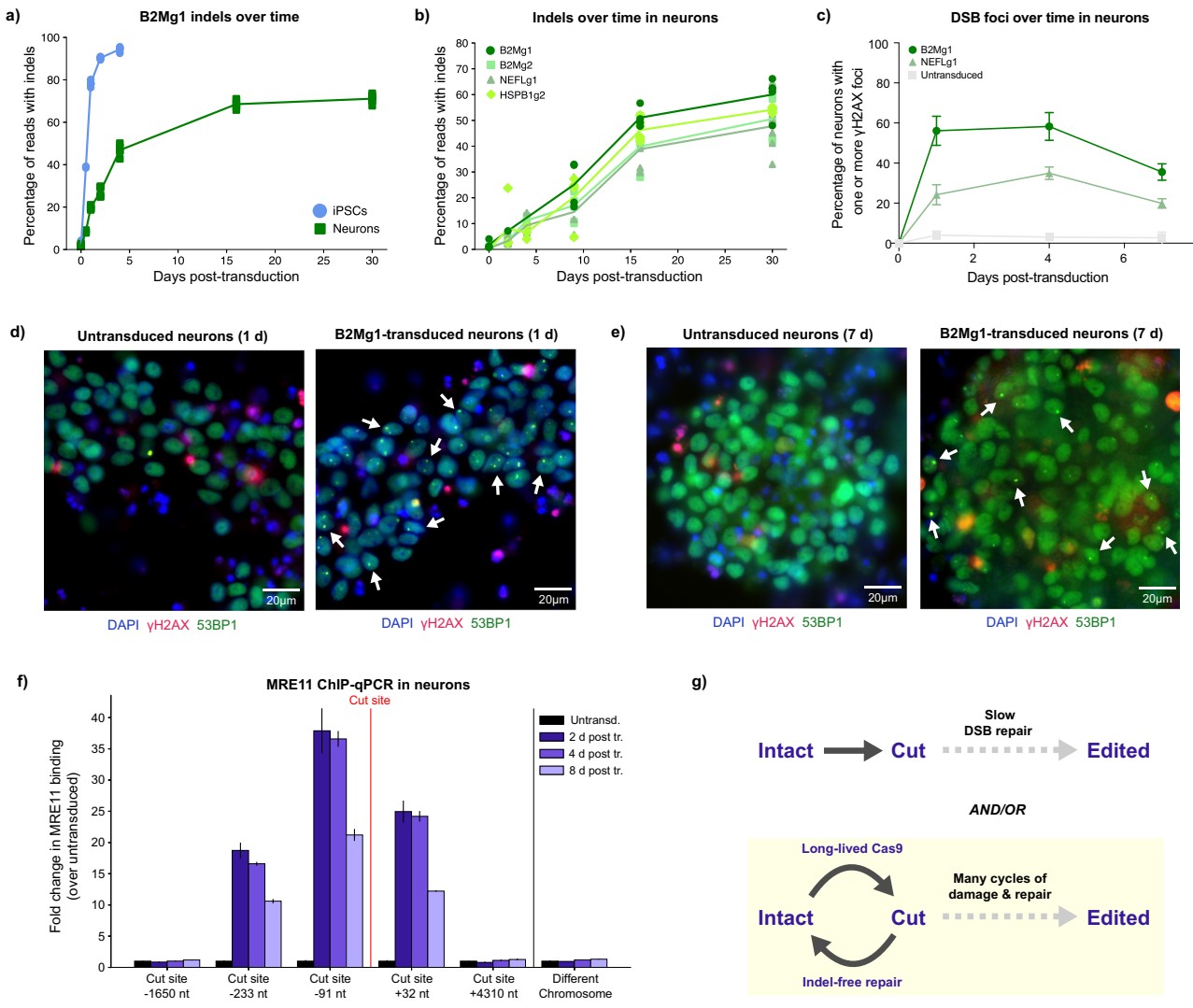

**Fig. 2 | Cas9-induced indels accumulate over a prolonged time span in neurons.**
**a** Cas9-induced indels accumulate more slowly in neurons than in genetically identical iPSCs. Dose: 2 μL B2Mg1 VLP (HIV) per 100 μL media. For (**a, b**) 6 replicate wells per condition transduced in parallel (some obscured by overlap); curves pass through the mean at each timepoint. CRISPResso2 analysis of amplicon-NGS.
**b** Several sgRNAs show weeks-long accumulation of indels in neurons. Dose: 1 μL VLP (FMLV) per 100 μL media. **c–e** Cas9-induced DSB foci (γH2AX+) remain detectable in neurons for at least 7 days post-transduction. Quantified in (**c**) by manual counting across *n* = 3 wells per condition; center points show means and error bars show SD. Representative ICC images of neurons 1 day (**d**) and 7 days (**e**) post-transduction, with age-matched untransduced neurons. Dose: 2 μL VLP (FMLV) per 100 μL media. See Supplementary Fig. 12 for unmerged/uncropped panels and full time course. ICC time course was conducted once; representative images chosen from 3 replicate wells per condition. **f** MRE11 is bound near the cut site in neurons for at least 8 days post-transduction. Dose: 2 μL B2Mg1 VLP (FMLV) per 100 μL media. Binding quantified by ChIP-qPCR, normalized for amplification efficiency and input chromatin. Average of 3 replicate reactions, normalized to untransduced control for each amplicon. Error bars show SD, centered at the mean. **g** Schematic: prolonged indel accumulation in neurons could be caused by neurons repairing DSBs more slowly, and/or by neurons undergoing more cycles of indel-free repair and re-cutting before edits arise. Our results do not rule out either model, but the early presence of post-repair products (Supplementary Fig. 13) and the surprising longevity of Cas9 protein in neurons (Supplementary Fig. 14) more strongly support the second model.

(Supplementary Fig. 12). This long-lived repair signal is consistent with the prolonged accumulation of indels in neurons. DSB foci in iPSCs cannot be compared over the same span, as proliferating cells replicate many times within a week, diluting any unresolved signal.

To more quantitatively measure this repair in neurons, we used chromatin immunoprecipitation with quantitative real-time PCR (ChIP-qPCR) to measure the binding of repair proteins Mre11 and γH2AX near the cut site, at several timepoints post-transduction. Mre11 binding in edited neurons was strongly detected within a few hundred bases of the cut site, and only in transduced samples (Fig. 2f), matching patterns seen in other cell types[42]. But intriguingly, Mre11 binding near the cut site remained strongly detected in neurons even 8 days post-transduction, decreasing by only ~50% between days 2 and 8.

As expected based on previous reports[42,43], γH2AX binding was much broader, with maximal signal detected several kilobases away from the cut site. Typically found between 2 and 30 kilobases away from the cut site, γH2AX is thought to coordinate longer-range interactions that facilitate repair, such as damage-induced cohesion to sister chromatids in dividing cells[43,44]. The role of γH2AX bound closer to the cut site remains unclear. Interestingly, while γH2AX binding farther from the cut site in our neurons decreased to background levels between days 2 and 8, γH2AX binding adjacent to the cut site only decreased by ~50% during this interval (Supplementary Fig. 13a). These week-long analyses cannot be performed in dividing cells like iPSCs, where one locus quickly becomes many due to replication.

Some Mre11/γH2AX binding at each timepoint can be attributed to DSBs that had already been repaired (with or without an indel). This is evidenced by their binding to an amplicon that spans across the cut site, and thus should only amplify if the cut was resealed (Supplementary Fig. 13b–d). Cut sites resealed without an indel can be repeatedly recut by any remaining Cas9 RNP, until an indel prevents subsequent Cas9 binding.

Altogether, multiple complementary approaches confirm that DSB repair signals at the target site persisted in neurons for much longer than expected, decreasing by only ~50% after 1 week post-delivery of Cas9 RNP. Two models could explain this prolonged timeline: postmitotic neurons might repair DSBs more slowly, or might undergo more cycles of repair and recutting until indels arise—or perhaps both (Fig. 2g).

### VLP-delivered Cas9 remains detectable in neurons for up to 30 days

We cannot rule out either model, but our evidence more strongly supports the latter model. First, while Cas9 in dividing cells is rapidly degraded and/or diluted by cell division, we found that Cas9 in non-dividing neurons is surprisingly long-lived. Following transient VLP delivery, Cas9 protein was undetectable in iPSCs after 8 days —but remained present in neurons even after 30 days (Supplementary Fig. 14a). Second, we found that this VLP-delivered Cas9 remains *functional* in neurons for at least a week. Even when we delivered Cas9-only VLP without any sgRNA, then delivered the sgRNA separately 8 days later, we still detected ~15% editing in neurons (Supplementary Fig. 14b). These findings are compatible with the model that DSBs in neurons tend to be repaired without indels, but long-lived Cas9 protein facilitates many cycles of damage and repair until an edit finally arises days or weeks later.

Either way, the longevity of Cas9 and Cas9-induced damage could have important consequences for the safety of genome editing in nondividing cells, in particular with regard to cytotoxicity, immunogenicity, and off-target edits.

### Cas9-VLPs elicit a striking transcription-level response in neurons

Given this unexpectedly prolonged time scale of editing, we reasoned that neuronal DNA repair might include transcription-level regulation, not only post-translational regulation. To test this, we used bulk RNA sequencing (RNAseq) to characterize differentially expressed genes (DEGs) in iPSCs and neurons transduced with Cas9-VLP, relative to untransduced cells (Supplementary Data 1). Unlike transduced iPSCs, transduced neurons exhibited a skewed transcriptional response, with far more genes upregulated than downregulated (Fig. 3a, b and Supplementary Fig. 15). The top 50 DEGs in neurons, all upregulated, were highly enriched for genes canonically associated with DNA repair and DNA replication (Fig. 3c).

This neuron-specific response was remarkably consistent regardless of the sgRNA target. In fact, only two genes were differentially expressed between B2Mg1-edited and NEFLg1-edited neurons: *B2M* and *NEFL*, respectively (Supplementary Fig. 15e). This confirms that the observed response is not locus-specific and is not driven by loss-of-function of either targeted gene. Surprisingly, over 75% of the DEGs in B2Mg1-edited or NEFLg1-edited neurons, relative to untransduced, were also shared with NTg1-treated neurons (Supplementary Fig. 16a, b). The top 50 DEGs shared by all three transduced neuron conditions were again highly enriched for DNA-interacting factors (Supplementary Fig. 16c, d). This suggests that Cas9-VLPs induce a strong transcription-level DNA repair response in neurons, some part of which may even be DSB-independent. Notably, this response was unique to neurons; DEGs were not enriched for DNA repair in any of the three transduced iPSC conditions (Supplementary Fig. 17). These DNA repair genes were already expressed at baseline in untransduced

iPSCs, whereas neurons only induced their expression upon Cas9-VLP transduction.

### Transduced neurons upregulate unexpected DNA repair genes

The most-significantly upregulated repair genes included many pathways thought to be inactive in nondividing cells, such as end resection-related pathways[22] (Supplementary Fig. 16d). They also included factors known to influence prime editing and base editing[9,10], suggesting this neuronal response could impact multiple editing modalities. Additionally, transduced neurons significantly upregulated factors that respond to R-loops, single-stranded DNA, and topological stresses (Supplementary Fig. 16d). This might explain why NTg1-Cas9 also induced a strong response: even if it does not cut, Cas9 still disrupts DNA, unwinding it, creating R-loops, and exposing single-stranded DNA[45,46].

Intriguingly, the most-upregulated genes in transduced neurons were particularly enriched for replication-related factors, such as cell cycle checkpoints and DNA synthesis during S phase (Fig. 3c and Supplementary Fig. 16). Neurons have long been postulated to partially re-enter the cell cycle following certain types of damage, through a process called endocycling which replicates DNA without necessarily completing mitosis[47–50]. Cas9-VLPs could have induced such a response in our neurons. It is also possible, however, that these repair factors are canonically annotated as replication-related because they have mostly been studied in dividing cells, where their role in repairing replication-induced damage eclipses any others. In nondividing cells, these factors' roles in responding to other types of DNA damage might be more visible. We investigated one of the strongest and most unexpected of these hits: *RRM2*.

### Transduced neurons non-canonically upregulate a subunit of ribonucleotide reductase

*RRM2* encodes a subunit of ribonucleotide reductase (RNR), the enzyme that produces deoxyribonucleoside triphosphates (dNTPs). RNR is functional when the catalytic subunit RRM1 binds one of two tightly regulated smaller subunits: RRM2 or RRM2B[51]. *RRM2* expression is canonically restricted to the S phase to produce dNTPs for replication, while *RRM2B* is canonically upregulated by p53 upon DNA damage to facilitate repair[52].

In iPSCs, each RNR subunit responded as expected: *RRM2B* was upregulated by the VLPs that induced DSBs, and *RRM2* was unaltered (Supplementary Fig. 18a–c). In contrast, the response in neurons was completely unexpected. *RRM2B* expression was not altered in any condition. Instead, the canonically S-phase-restricted *RRM2* was one of the most significantly upregulated genes transcriptome-wide, in every transduced neuron condition —including those treated with non-targeting (NTg1) Cas9-VLP (Supplementary Fig. 18d–f). Therefore, we investigated how much of this upregulation was induced by DNA damage specifically, vs Cas9 protein, or even VLPs themselves.

### This transcriptional response is amplified by, but not specific to, Cas9-induced DSBs

To test this, we performed bulk RNAseq on neurons transduced with VLPs delivering Cas9-B2Mg1, Cas9-NTg1, dCas9-B2Mg1, or GFP (Supplementary Data 2). Unbiased clustering showed that neurons treated with GFP-VLPs were distinct from those treated with Cas9-VLPs, but they still upregulated repair factors including *RRM2* (Supplementary Fig. 19a–c). This suggests that upregulation of DNA repair factors in neurons can be induced by VLPs themselves, independent of Cas9. However, the presence of Cas9 DSBs significantly amplified this neuronal response. Compared to the non-targeting Cas9-NTg1, Cas9-B2Mg1 induced over 2-fold higher *RRM2* expression, and significantly upregulated pathways canonically associated with replication (Supplementary Fig. 19d, e). *RRM2* expression was also more than 2-fold

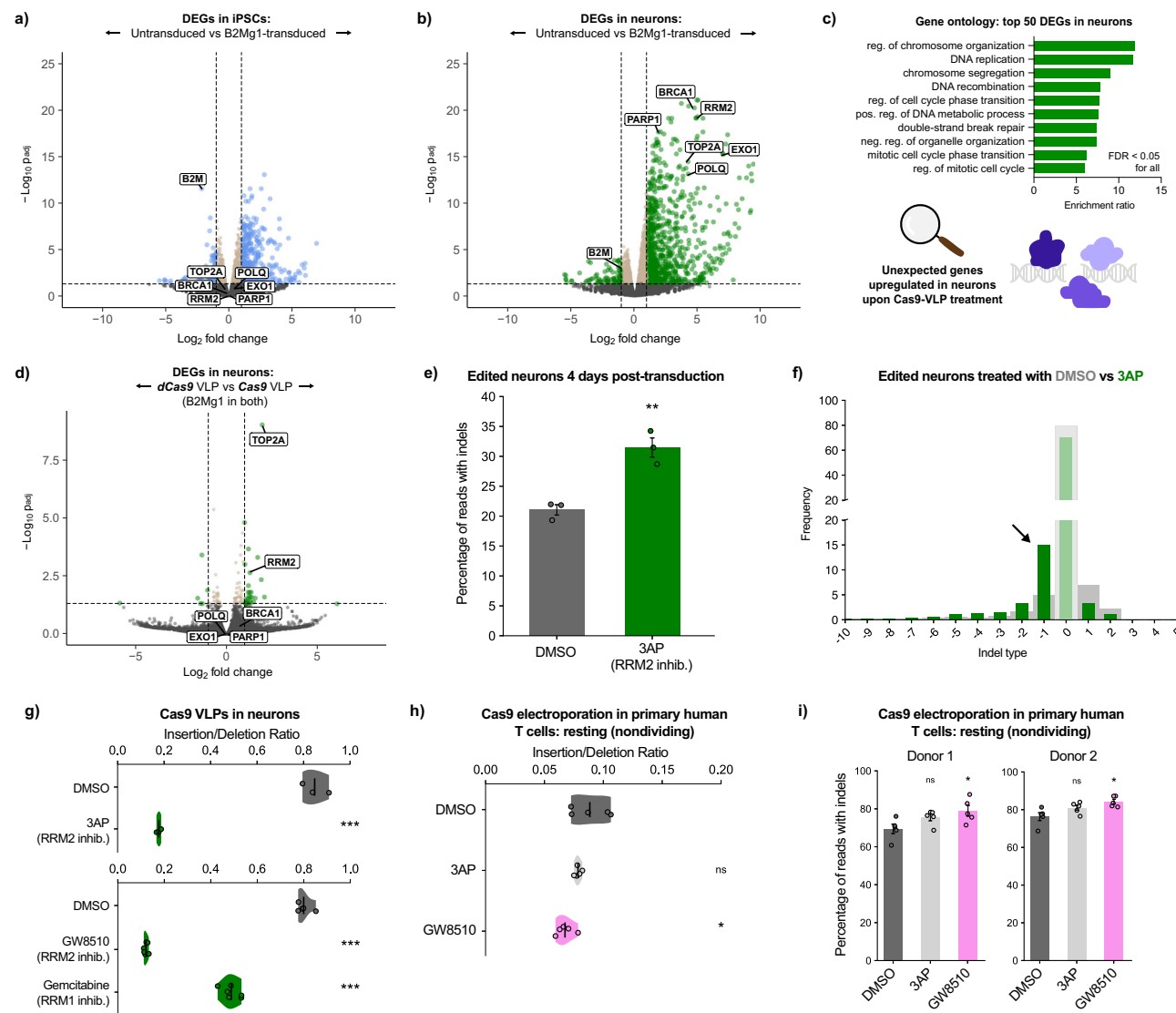

**Fig. 3 | Neuronal response to Cas9 reveals unexpected factors that influence editing outcomes in nondividing cells.** Neurons (**b**), but not iPSCs (**a**), dramatically upregulate transcription of DNA repair factors upon Cas9-VLP transduction. For **a**–**d**: dashed lines show cutoffs for significance ($p_{adj} < 0.05$) and effect size (fold-change > 2 or < 0.5). Statistical tests detailed in "Methods" section under "Bulk RNAseq". For **a**–**c**, dose: 1 μL HIV VLP per 20,000 cells. **c** The most significantly altered DEGs in Cas9-VLP-treated neurons are highly enriched for factors canonically associated with DNA replication/repair. **d** RRM2 is more strongly upregulated in neurons treated with Cas9-VLP compared to dCas9-VLP. Dose: 2 μL FMLV VLP per 20,000 cells. **e** Inhibiting RRM2 yields a ~50% increase in neuron editing efficiency, within 4 days post-transduction. Dose: 1 μL B2Mg1 VLP (FMLV) per 20,000 cells in 100 μL media. Error bars show SEM. One-factor ANOVA, **$p < 0.005$. For **e**, **f**: $n = 3$ replicate wells transduced. For e-i: CRISPResso2 analysis of amplicon-NGS. **f** RRM2 inhibition shifts neuron indels from insertions toward deletions, and triples the frequency of 1-base deletions at 4 days post-transduction. Thick gray bars are DMSO condition; thin green bars are 3AP. **g** Inhibition of RRM2 or RRM1 consistently shifts neuron indels from insertions toward deletions. Dose: 2 μL B2Mg1 VLP (FMLV) per 20,000 cells in 100 μL media. $n = 3$ (top) or 6 (bottom) replicate wells transduced. **h** RRM2 inhibition also shifts indels from insertions toward deletions in resting (nondividing) primary human T cells. For **h**, **i**: $n = 5$ replicate electroporations with 6.25 pmol Cas9 RNP (B2Mg1), harvested 4 days post-electroporation. **i** RRM2 inhibition boosts total indel efficiency in resting primary human T cells from two independent donors. For **g**–**i**: Error bars show SEM. One-factor ANOVA with Tukey's multiple comparison test, *$p < 0.05$, ***$p < 0.0005$, ns not significant.

higher in neurons treated with Cas9-B2Mg1 than with dCas9-B2Mg1 (Fig. 3d and Supplementary Fig. 19f, g).

These results indicate that neurons' transcriptional response to VLPs is amplified by, but not specific to, Cas9-induced DNA damage. Regardless of what stresses are capable of inducing this response, its end result is the significant upregulation of non-canonical DNA repair factors in neurons. We hypothesized that this unexpected shift in the DNA repair landscape could impact CRISPR editing. For example, in NHEJ processing where polymerase filling-in competes with other pathways[34] (Supplementary Fig. 1), this non-canonical RNR activation could bias the outcome by increasing nucleotide availability.

## Inhibiting RNR influences editing outcomes in neurons

Based on these results, we tested whether inhibiting RNR affected Cas9 editing outcomes. We treated neurons with triapine (3AP), a small molecule inhibitor of RRM2[53–56], while delivering Cas9-VLPs targeting B2Mg1. Excitingly, 3AP treatment led to a ~50% increase in total indels, at only four days post-transduction (Fig. 3e). This increase in indels came almost exclusively from boosting deletions, at the expense of insertions and indel-free repair (Fig. 3f). In fact, 3AP co-treatment led to a ~3-fold increase in single-base deletions specifically, tilting the distribution toward one predictable outcome.

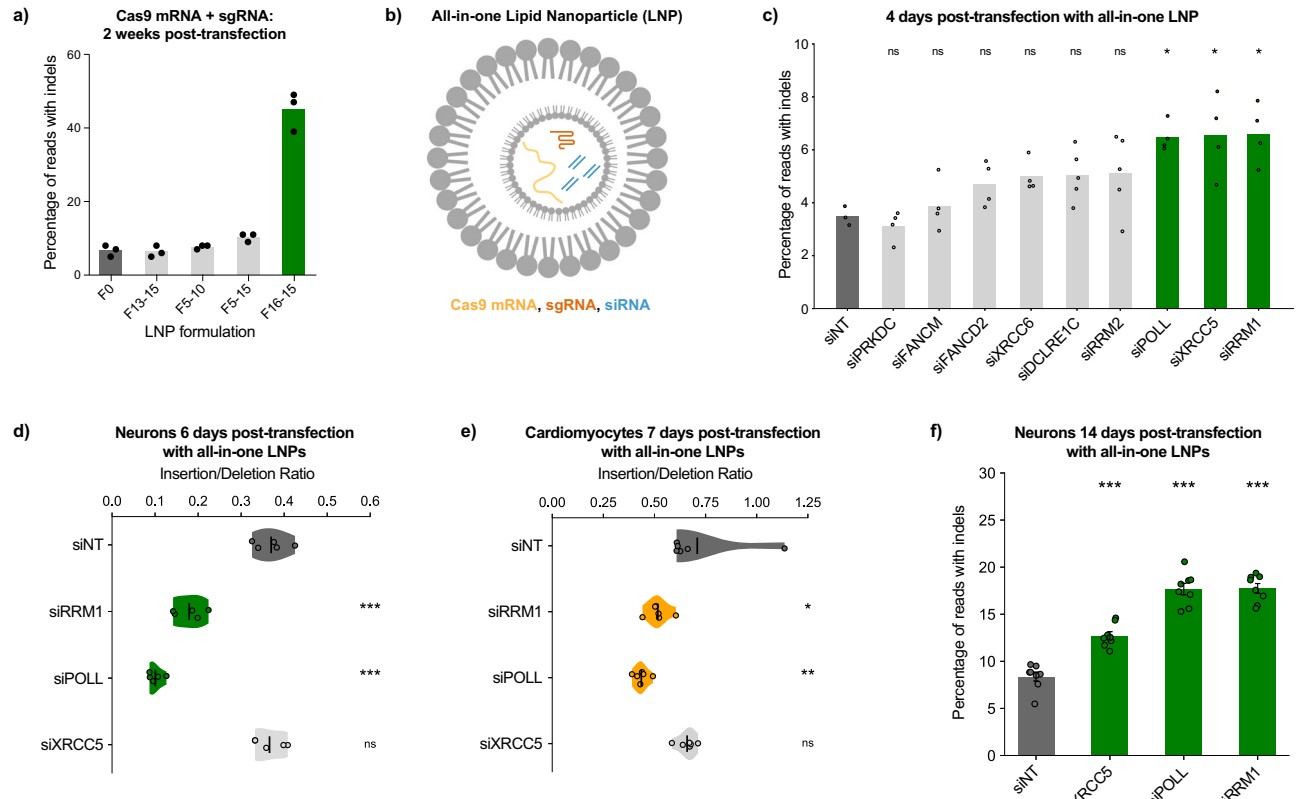

**Fig. 4 | All-in-one particles deliver Cas9 and sgRNA while simultaneously manipulating DNA repair factors. a** LNP formulation F16-15 effectively delivers Cas9 mRNA and sgRNA to human iPSC-derived neurons. "F0" is DLin-MC3-DMA. $n = 3$ replicate wells transfected. For **a–f**: 125 ng total RNA per 20,000 cells in 100 μL media, targeting B2Mg1. For **a**: Synthego ICEv2 analysis, 2 weeks post-transfection. For **c–f**: CRISPResso2 analysis of amplicon-NGS, averaged across $n = 5$ (**c**), 8 (**d**), 6 (**e**), or 8 (**f**) replicate wells per condition transfected in parallel. **b** Schematic illustrating all-in-one LNPs that encapsulate Cas9 mRNA (yellow) and sgRNA (orange), along with siRNAs (blue) against a repair gene of interest. **c** All-in-one LNPs reveal additional targets that increase editing efficiency at 4 days post-

transfection. One-Factor ANOVA with Tukey's multiple comparison test, for each condition vs siNT: $^*p < 0.05$, ns not significant. In both neurons (**d**) and cardio-myocytes (**e**), all-in-one LNPs knocking down RRM1 or POLL each phenocopy drug inhibition of RNR: shifting indels strongly from insertions toward deletions. Inhibiting XRCC5 boosts both insertions and deletions (see Supplementary Fig. 24). One-factor ANOVA with Tukey's multiple comparison test. Each condition vs siNT, $^*p < 0.05$, $^{**}p < 0.005$, $^{***}p < 0.0005$, ns not significant. **f** Knockdown of XRCC5, POLL, or RRM1 significantly increases total editing efficiency in neurons at 2 weeks post-transfection. One-factor ANOVA with Tukey's multiple comparison test. Each condition vs siNT, $^{***}p < 0.0005$. Error bars show SEM.

Two other RNR-inhibiting drugs had the same effect as 3AP on B2Mg1 editing in neurons: GW8510, which also inhibits RRM2[54,57,58], and gemcitabine, which inhibits its obligate binding partner RRM1[59,60]. Both drugs increased total indel frequency, and preferentially boosted deletions, affecting both the efficiency and precision of gene inactivation (Fig. 3g and Supplementary Fig. 20). 3AP and gemcitabine have already been used in clinical trials for other applications[61–64]. Depending on the toxicity to dividing cells, future studies could explore their clinical relevance for enhancing therapeutic editing outcomes.

Expecting that different cut sites may differ in scission profiles and repair dependencies[65–67], we also tested the effect of RNR inhibition on three other sgRNAs. Inhibiting RNR had the same effect for B2Mg2 and NEFLg1 as it did for B2Mg1: increasing total editing efficiency, and shifting indels from insertions toward deletions (Supplementary Fig. 21). For HSPB1g2 however, RNR inhibition decreased total indels (Supplementary Fig. 21). This is consistent with the intrinsic indel distribution of HSPB1g2, which appears impermissible to the deletions that RNR inhibition typically boosts. Therefore, inhibiting RNR cannot be generalized as a method to increase indel efficiency for all sgRNAs. Rather, RNR inhibition influences editing outcomes in an sgRNA-dependent manner.

## Inhibiting RNR influences editing outcomes in primary human T cells

To test whether these effects were simply an artifact of our iPSC-derived neuron model or VLP-delivered Cas9, we next investigated how RNR inhibition affected *primary* human T cells after electroporation with Cas9 RNP. Cas9-based inactivation of *B2M* in engineered T cells is being explored as a clinical strategy to reduce the immunogenicity of allogeneic T cell therapies[68]. It has recently been reported that editing these T cells in the non-activated state reduces genotoxicity and chromosome loss[69].

In our resting (nondividing) primary T cells, inhibiting RNR reproduced the same effect on indel outcomes as seen in our neurons. *RRM2* inhibition by GW8510 significantly decreased the insertion/deletion ratio, across two different sgRNAs and two different human donors (Fig. 3h and Supplementary Fig. 22). Furthermore, across both human donors, *RRM2* inhibition significantly boosted overall editing efficiency for B2Mg1 in resting primary T cells (Fig. 3i and Supplementary Fig. 22). This was not true for the more insertion-biased B2Mg3 (Supplementary Fig. 22), just as the more insertion-biased HSPB1g2 was the exception in neurons. Together, these results corroborate that RNR inhibition consistently shifts indels from insertions toward deletions in clinically relevant nondividing cells, and that this shift can increase total editing efficiency in an sgRNA-dependent manner.

Overall, deciphering how clinically relevant cells respond to Cas9 unveiled unexpected DNA repair factors that influence editing outcomes. Identifying these upregulated genes highlighted many potential targets for manipulating repair. Our RNR results demonstrate that modulating these factors can reveal which outcomes they affect and can help optimize the editing outcome for a given sgRNA of interest.

## Prolonged editing window allows manipulation of repair factors at RNA level

When modulating DNA repair factors to optimize editing outcomes, a major barrier is that not all proteins are druggable. Since neurons activated DNA repair factors at the transcriptional level as well, and had a long window of days or weeks for completing repair, we reasoned that manipulating repair factors at the RNA level−rather than the protein level−may also be sufficient to influence neuron indels. If true, this would enable modulation of any DNA repair factor, not only the druggable ones. To test this idea, we used short interfering RNAs (siRNAs) to inhibit DNA repair genes of interest, delivered along with Cas9 mRNA and sgRNA inside lipid nanoparticles (LNPs), a delivery vehicle well-suited for an all-RNA cargo.

First we selected 5 LNP formulations that have previously been used in mouse neurons, including the commercially available DLin-MC3-DMA ("F0"), and tested them in human iPSC-derived neurons. Most of these formulations delivered GFP mRNA to our neurons effectively (Supplementary Fig. 23a), albeit with visible toxicity. For delivery of Cas9 mRNA and sgRNA however, formulation F16-15 was by far the most effective (Fig. 4a and Supplementary Fig. 23b). While median GFP fluorescence per neuron was over 4-fold higher with F0 than with F16-15, Cas9 editing was over 2-fold more efficient with F16-15 than with F0 (Supplementary Fig. 23c, d). This demonstrates that the optimal LNP formulation for a given cell type can vary based on the type of cargo being delivered.

Next, using F16-15, we designed "all-in-one LNPs" which co-encapsulate 5 distinct RNA species: Cas9 mRNA, sgRNA, and a mix of 3 siRNAs knocking down a DNA repair gene of interest (Fig. 4b). To evaluate whether this RNA-level inhibition of repair genes can influence indel outcomes, we explored siRNA knockdown of RNR subunits. Since *RRM2* is not expressed in neurons until after Cas9 exposure−yet the siRNA is active before the Cas9 mRNA gets translated−we inhibited its obligate binding partner *RRM1*, which is more highly expressed at baseline. This siRNA treatment during B2Mg1 editing phenocopied small molecule inhibition of *RRM1/2*. Two weeks post-transfection, siRNA inhibition of *RRM1* shifted indels from insertions toward deletions, increasing the frequency of single-base deletions by ~75% (Supplementary Fig. 24a). Therefore, these all-in-one LNPs allowed us to deliver editing reagents to neurons while simultaneously influencing the repair outcome with RNA interference (RNAi). This co-packaging strategy might be safer than systemically delivering drugs that are toxic to dividing cells, and it also allows us to target repair factors even when they are not druggable.

## All-in-one particles enable arrayed screening for repair factors that influence editing outcomes

We then demonstrated the use of all-in-one LNPs for arrayed screening to optimize genome editing. We transfected neurons with all-in-one particles targeting a set of additional DSB repair factors, several of which are not reliably targetable by small molecule drugs. Aiming to identify perturbations that accelerate editing, we assessed editing at an earlier time point of 4 days post-transfection, before indels had plateaued. Knockdowns of *RRM1*, *POLL*, and *XRCC5* significantly increased total B2Mg1 indels by ~80% relative to non-targeting siRNA (siNT) (Fig. 4c). *POLL* encodes the polymerase that likely performs filling-in synthesis during NHEJ processing[34], using the dNTPs produced by RNR. And *XRCC5* (Ku80) is one of the key factors involved in end

protection to promote indel-free cNHEJ[34,35]. Our model is that these interventions disrupted indel-free repair of the B2Mg1 cut site and directed the repair outcome toward indels instead, thus accelerating gene inactivation. Subsequent validation confirmed that each of these 3 hits increased editing efficiency at both early and late timepoints in neurons (Supplementary Fig. 24b, c).

## Hits from arrayed screen alter CRISPR repair outcomes in neurons and cardiomyocytes

Knocking down *RRM1* or *POLL* boosted deletions preferentially, significantly decreasing the insertion/deletion ratio (Fig. 4d and Supplementary Fig. 24d–f). Knocking down *XRCC5* boosted both insertions and deletions, without significantly altering the insertion/deletion ratio. Excitingly, we also reproduced these effects using the same F16-15 all-in-one LNPs in nondividing cardiomyocytes: siRRM1 and siPOLL shifted indels from insertions toward deletions, while siXRCC5 increased both insertions and deletions (Fig. 4e and Supplementary Fig. 24g–j). Therefore, our all-in-one LNPs helped reveal DNA repair mechanisms that are shared between postmitotic neurons and cardiomyocytes. In neurons, perturbing those mechanisms provided 1.5-fold to 2-fold increases in total editing at 14 days post-transfection (Fig. 4f).

This arrayed screening platform could be used to find optimal repair modifications for any sgRNA in a cell type of interest, simply by encapsulating different sgRNAs and siRNAs inside the all-in-one particles. By gaining better control over which repair pathways a cell utilizes, we could help minimize the risk of persistent/unresolved DSBs and improve the precision and predictability of genome editing outcomes.

## Discussion

Our results highlight the importance of testing genome editing therapies in the appropriate cell type. Through advances in human cell models and nonviral CRISPR delivery tools, we were able to model transient CRISPR editing of human genome sequences in human postmitotic cells. We found that neurons' distinct responses to Cas9-induced DNA damage led to dramatically different repair outcomes and weeks-long accumulation of edits, potentially impacting the safety of CRISPR therapies. Fortunately, we also identified strategies to manipulate neuronal DNA repair pathways to improve the efficiency and precision of genome editing.

Investigating the neuronal response to Cas9 revealed that neurons expressed many DNA repair genes only *after* acute damage occurred. Therefore, expression levels of repair genes in unperturbed cells should not be used as a proxy for which repair pathways are accessible. For example, *RRM2* is canonically considered completely inactive in nondividing cells, yet by inhibiting it we impacted both the efficiency and precision of Cas9 indels in neurons. Using both chemical and genetic inhibition, we validated RNR's effect on insertion/deletion ratio across three nondividing human cell types (including primary cells), and across three different modes of Cas9 delivery. These perturbations also helped narrow the distribution of resulting indel types, making neuron editing outcomes more precise and predictable. However, we found that the optimal DNA repair perturbations for one sgRNA cannot necessarily be generalized to other sgRNAs with different scission profiles.

These insights also helped turn neurons' slow indel accumulation from a challenge into an opportunity. Since the neuronal repair response was detectable for many days and involved transcription-level upregulation, we devised all-in-one particles to deliver Cas9 while simultaneously manipulating the repair process via RNAi. Compared to drug inhibition, this strategy greatly expanded how many factors we can now target. As a proof-of-concept, we used this all-in-one screening platform to find repair modifications that accelerate indels for a given sgRNA of interest. Future studies could use our tools to optimize

the safety and efficacy of DSB-independent editing modalities in neurons as well.

Overall, by studying how nondividing cells repair Cas9-induced DNA damage, we discovered new strategies to influence genome editing outcomes. By reproducing key results across multiple clinically relevant types, we showed that these findings could impact future genome editing therapies in neurons and cardiomyocytes, as well as imminent therapies in T cells.

Our experimental approach benefited from several key strengths. First, using iPSCs instead of transformed cell lines allowed us to model DNA repair in karyotypically normal cells. Second, comparing neurons to their isogenic iPSCs allowed us to evaluate CRISPR editing in dividing vs nondividing cells without confounding factors such as genetic background. Third, since iPSC-derived neurons share the genotypes and even some phenotypes of the human donors, this same platform can be used to test and optimize a genome editing therapy in a patient's own iPSC-derived neurons. Fourth, we used nonviral particles to deliver transient Cas9 RNP or mRNA, rather than viral vectors delivering DNA-encoded Cas9. This avoided indefinite Cas9 expression that would have obscured the prolonged editing time course, and avoided exogenous DNA episomes that could be aberrantly integrated into long-lived DSBs. Nonviral delivery to postmitotic human neurons has long been a challenge for the field; utilizing recent advances in VLP and LNP technology to overcome this barrier was crucial to studying neuronal DNA repair accurately.

Our approach also had several limitations, which could be addressed by future follow-up studies. First, it is very likely that other untested sgRNAs, and/or other nucleases, will have different DSB repair dependencies than the ones revealed by this study. Future studies pairing our platform with higher-throughput methods such as CRISPR interference (CRISPRi) screening could identify optimal repair modifications for particular sgRNAs of interest, and study neuronal responses to other genome editing enzymes. Second, the RNAseq showing upregulation of DNA repair factors in VLP-treated neurons was reproduced twice in neurons from the WTC-NGN2-CRISPRi background, but curiously did not reproduce in WTC-NGN2 neurons lacking CRISPRi. It is uncertain whether the observed DEGs between transduced and untransduced neurons could have somehow been affected by the presence of dCas9 (although it was present in both conditions) or by cell line variability. Regardless, the other experiments in this study, including subsequent validation experiments like RNR inhibition, were performed in cells without CRISPRi – avoiding this potential confounder. Third, while our all-in-one LNP platform is a useful genetic tool for exploring DNA repair biology, its therapeutic utility is currently more limited. Since Cas9 delivery remains a rate-limiting step, overall editing with LNPs was still higher when siRNA was replaced entirely with more Cas9 mRNA and sgRNA instead. This could be improved by further optimizing delivery formulations and ratios, perturbation strategies, and the timing of perturbations relative to editing. Finally, even though we used postmitotic human neurons, it is unknown how our findings will translate to aged/diseased neurons in patients, or nonhuman neurons in animal models. Future studies could investigate the timing and repair of edits in rodent/primate neurons, and potentially in ex vivo primary human neurons.

In summary, examining how postmitotic neurons respond to CRISPR perturbations uncovered new considerations for safety and efficacy, and new avenues for controlling CRISPR repair outcomes. The genome modification toolkit contains several tools to perturb DNA, but we are just beginning to develop tools that ensure proper repair. Those tools will be crucial for unlocking the full potential of therapeutic genome editing.

## Methods

### iPSC maintenance

iPSCs were cultured on matrigel-coated 10 cm plates at 37 °C, 85% humidity, and 5% $CO_2$. iPSCs were fed with mTeSR Plus media (Stem-Cell Tech #100-0276) every other day. Optionally, if fed with double the feeding volume of mTeSR Plus one day after passaging, iPSC media could be left unchanged for two days. Upon reaching 80% confluence, iPSCs were passaged 1:10 or 1:20 and treated with 10 µM ROCK inhibitor (Y-27632 dihydrochloride, e.g., Tocris #1254). For maintenance, ReLeSR (StemCell Tech #100-0483) was used to passage iPSCs as small colonies roughly twice per week. For seeding specific numbers of cells for experiments, Accutase (StemCell Tech #07920) was used to replate iPSCs as single cells after counting.

Cell lines were routinely verified as mycoplasma negative throughout the study. WTC-NGN2 iPSCs were used for all experiments except RNAseq, which used WTC-NGN2-CRISPRi. WTC-NGN2 is the WTC11 iPSC line (Coriell GM25256, male) with the dox-inducible NGN2 differentiation cassette integrated in the AAVS1 locus.

### Neuron differentiation

Neurons were derived from WTC-NGN2 iPSCs following a differentiation protocol adapted from Tian et al., Neuron (2019), PMID: 31422865. Note however, that instead of naming the first day of differentiation Day −3, we name it Day 0. Refer to Supplementary Data 3 for our adapted differentiation protocol and spreadsheet to aid in calculations.

On Day 3 of differentiation, neurons were seeded onto PDL-coated culture plates (e.g., Corning #354640, #356414, #356413, #354469): 96-well plates for editing assays, 24-well plates for flow cytometry assays, 6-well plates for RNA assays, or 10 cm plates for ChIP-qPCR. Critically, to maintain neuron viability and reduce media evaporation, we added PBS to the unused wells surrounding cell-seeded wells, especially in 96-well plates. Additionally, to reduce neuronal peeling, for media changes from Day 10 onward, we typically removed only half of the existing media volume per well and added a full feeding volume—except when adding VLPs/LNPs/drugs, for which full media changes were used to accurately control concentrations.

In 96-well plate format, each well contained ~20,000 cells and was treated with 100 µL of VLP- or LNP- containing media. In larger plate formats, these ratios were scaled up proportionally. Note: to transduce 20,000 iPSCs on the same day as the neurons, 10,000 iPSCs were seeded per well 1 day prior, or 5000 iPSCs per well were seeded 2 days prior. Whereas for neurons, 20,000 were seeded on Day 3 of differentiation.

### VLP production and transduction

For HIV-based VLPs (also known as enveloped delivery vehicles or EDVs), we followed the protocols previously described in Hamilton et al., Nat Biotechnol (2024), PMID: 38212493. For FMLV-based VLPs, refer to Supplementary Data 4 for our full production protocol and calculations; the necessary plasmids have been deposited to Addgene (#225959, #225960, #225961, #225962, #225963).

For both particle types, each "batch" of VLPs consisted of six 10 cm dishes of transfected HEK 293FTs. 44–48 h post-transfection, each batch's supernatant was harvested, purified using Lenti-X Concentrator (Takara #631231), and concentrated into 200 µL of OptiMEM (e.g., Gibco #31985062). Dosage: VLP doses listed in figure captions (either 1 or 2 µL as specified) refer to how many µL of this concentrated VLP solution were added per 100 µL of cell culture media, for transduction.

For DSB imaging experiments, transduction was done at Day 14 of differentiation. For all other experiments, transduction was done at Day 17+.

## LNP production and transfection

Lipid mixtures for LNPs were prepared according to previously published procedures (https://doi.org/10.1021/acs.biochem.3c00371). Briefly, stock solutions (10 mg/mL) of MC3 (MedKoo, cat. # 555308), DOPE (Avanti Polar Lipids, cat. # 850725), cholesterol (Sigma–Aldrich, cat. # C8667), and DMG-PEG (Avanti Polar Lipids, cat. # 880151) were individually dissolved in ethanol, while GL67 (N4-Cholesteryl-Spermine HCl Salt, Avanti Polar Lipids, cat. #890893) was dissolved in DMSO. These lipid stock solutions were stored at −30 °C until use. Prior to LNP formation, the lipid solutions were thawed on ice and vortexed as needed. The cholesterol solution was warmed at 40–50 °C to dissolve any crystals that formed during cold storage. Subsequently, MC3, DOPE, cholesterol, DMG-PEG, and GL67 lipids were mixed in molar ratios of 30.8:20.8:32.2:1.2:15, respectively. RNA (1 µg/µL) was dissolved in 200 mM citrate buffer (pH 4), aliquotted, and stored at −80 °C.

Shortly before use, RNA was diluted to 375 ng/µL, then combined with the lipid solution at a 3:1 volume ratio of aqueous phase to lipids. The resulting LNP mRNA complexes were gently vortexed or triturated, and incubated at room temperature for 5–10 min. Finally, the LNPs were mixed with the appropriate volume of cell culture media, and added to cells during a full media change. Dosage: we added 300 µL of cell culture media per 4 µL of LNP solution (1 µL of which is lipids). This resulted in an RNA dosage of 125 ng total RNA per 100 µL cell culture media.

Chemically modified GFP mRNA (cat. # L-7201) and Cas9 mRNA (cat. # L-7206) were purchased from TriLink. Chemically modified sgRNAs were ordered from IDT; refer to Supplementary Data 5 for ordering instructions and resuspension instructions. For siRNAs, TriFECTa DsiRNA kits were ordered from IDT; we used the default TriFECTa kit targeting each gene of interest (cat. #s: hs.Ri.PRKDC.13, hs.Ri.RRM1.13, hs.Ri.RRM2.13, hs.Ri.POLL.13, hs.Ri.XRCC5.13, hs.Ri.XRCC6.13, hs.Ri.DCLRE1C.13, hs.Ri.FANCM.13, hs.Ri.FANCD2.13).

When delivering Cas9 mRNA and sgRNA, total RNA mass inside the particle was split 1:1 between Cas9 mRNA and sgRNA. For all-in-one particles co-delivering siRNAs as well, siRNAs were included such that the final concentration of "siRNA mixture" in wells was 10 nM. The amount of Cas9 mRNA + sgRNA was reduced proportionally to keep the total RNA concentration unchanged. Each "siRNA mixture" is a 1:1:1 mixture of the 3 individual siRNAs contained in the IDT TriFECTa kit for a given gene of interest—or, for siNT, it is the TriFECTa kit's included non-targeting negative control siRNA (labeled NC-1), at the equivalent concentration of total siRNA.

For all LNP experiments in neurons, transduction was done at Day 17+ of differentiation.

## Genomic DNA extraction, NGS, and editing analysis

All gDNA for editing experiments was harvested using QuickExtract (#QE09050). After removing cell culture media, 25 µL of QuickExtract was dispensed into each well, and cells were scraped and collected into PCR tube strips (eg Genesee #27-125) or tear-away PCR plates (4titude #4ti-0750/TA). Samples were incubated in a thermocycler for 65 °C for 20 min, then 98 °C for 20 min. Extracted gDNA was then stored at −20 °C for short term storage or −80 °C for long term storage.

PCR amplification was done with NEB Q5 master mix (NEB #M0492) and 34 cycles of amplification. Amplicons were then purified using PCR cleanup beads from the UC Berkeley DNA Sequencing Facility, with at least 15 min of post-ethanol drying time, and eluted in 30–40 µL of DEPC-treated water. Finally, purified samples were submitted to the UC Berkeley/IGI NGS Core for sequencing via Illumina iSeq (2 × 150), with 20,000 reads per sample. We processed the resulting sequencing files in Geneious Prime, then used CRISPResso2 (DOI:10.1038/s41587-019-0032-3. PMID: 30809026) to analyze editing outcomes.

For sgRNA spacer sequences and amplicon-NGS primers, refer to Supplementary Data 6. For LNP production, chemically modified synthetic sgRNAs were ordered from IDT as described above. For VLP production, spacer sequences were ordered as overlapping top and bottom oligos from IDT, annealed, and ligated into backbone plasmid pCF142_U6-sgRNA (deposited to Addgene, #225960) for expression.

## Drug treatments

Small molecules were resuspended as advised by the manufacturers. Stock concentrations were then prepared at 1000× the desired concentration: refer to Supplementary Data 7 for stock and final concentrations, as well as catalog numbers. Desired final concentrations were determined by measuring neuronal viability (via PrestoBlue) after escalating drug doses, as shown in Supplementary Fig. 20. Per manufacturer's suggestions, gemcitabine was resuspended in cell culture-grade water; other drugs were resuspended in DMSO.

Drug treatment during Cas9-VLP editing experiments was begun one day prior to transduction. To reduce neuronal peeling from excessive media changes, we did not remove any media during the one-day pre-treatment. Instead, we added one feeding volume on top of the existing media, with double the desired drug concentration, to achieve the desired final concentration in the well. The following day (the day of transduction), we performed a full media change, adding media mixed with the desired final concentration of drug and desired volume of VLPs.

## T cell isolation, culture, and editing

Human peripheral blood Leukopaks were procured from healthy donors (StemCell). Peripheral blood mononuclear cells were purified via serial centrifugation. Bulk T cells were isolated via immunomagnetic negative selection using the EasySep Human T Cell Isolation Cocktail (StemCell) and EasySep Dextran Rapid Spheres (StemCell). When specified, T cells were activated with CTS Dynabeads CD3/CD28 (Gibco) and cultured in T175 flasks. CTS Dynabeads were removed from activated cells before electroporation via magnetic selection. Activated T cells were split 1:2 every other day with complete X-vivo media (Lonza) with 5 ng/mL IL-7 and 5 ng/mL IL-15. Resting T cells were replenished with fresh X-vivo media (Lonza) with 5 ng/mL IL-7 every other day. At each timepoint, genomic DNA was harvested using QuickExtract DNA Extraction Solution (LGC Biosearch Technologies).

For electroporation: crRNAs (Dharmacon) and tracrRNA (Dharmacon) were both diluted to 160 µM with nuclease-free duplex buffer (IDT) and annealed at 37 °C for 15 min to form sgRNA. Cas9 RNPs were formed by combining the sgRNA and 40 µM Cas9-NLS (UC Berkeley/QB3 MacroLab) at a molar ratio of 2:1. Cas9 RNPs were first diluted to the appropriate working concentrations (1, 5, or 40 µM) with nuclease-free duplex buffer, to yield the specified final doses of Cas9 (1.25, 6.25, or 50 pmol) in each nucleofection well. Electroporation was performed using a 96-well format 4D-nucleofector (Lonza) with 2e6 cells per well. T cells were electroporated with P3 buffer (Lonza) using pulse code EH-115. Resting T cells were then resuspended in complete X-Vivo media (Lonza) with 5 ng/mL IL-7. Activated T cells were then resuspended in complete X-Vivo media with 5 ng/mL IL-7 and 5 ng/mL IL-15. When specified, RNR inhibitor drugs were added to appropriate T cell cultures 24 h prior to electroporation, omitted from the nucleofection buffer and rescue media, then added back into the complete media following electroporation. Drugs were then omitted when the cells were split 48 h post-electroporation.

## PrestoBlue viability assay

We performed a full media change on neurons, adding in 10% PrestoBlue Cell Viability Reagent (Invitrogen #A13261) by volume, and incubated the cells at 37 °C for 1–2 h prior to analysis. In 3 control wells with no cells, media with 10% PrestoBlue reagent was added to gauge background fluorescence. After this incubation, the plate was read on a

Molecular Devices SpectraMax plate reader and analyzed using the SoftMax software. The average background fluorescence from the control wells was subtracted from all experimental values.

## DSB marker staining and imaging

To stain and image markers of DSB repair (γH2AX and 53BP1), neurons were first fixed with 4% PFA for 15 min at room temperature (RT), washed with PBS, and permeabilized with 0.5% Triton X-100 in PBS for 5 min at RT. Neurons were then washed with PBS, and incubated with blocking buffer (1% BSA and 0.1% Triton X-100 in PBS) for 1 h at RT. Then, neurons were incubated with the following buffers at RT, with two PBS washes after each incubation: primary antibody solution for 1 h (Mouse Anti-phospho-Histone H2A.X Ser139 Antibody, clone JBW301, Sigma #05-636, 1:4000 diluted in blocking buffer; Rabbit Anti-53BP1 Antibody, Novus #100-305, 1:1000 diluted), secondary antibody solution for 1 h (Goat anti-Mouse IgG H + L 568, Invitrogen #A-11031; Goat anti-Rabbit IgG H + L 488, Invitrogen #A-11034; both 1:1000 diluted in blocking buffer), then DAPI for 2 min (1:1000 diluted in PBS, Thermo #62248). Finally, PBS was added to each well, and plates were stored foiled at 4 °C until ready to image.

Higher magnification DSB imaging was performed on a Nikon spinning disk confocal microscope, equipped with a Yokogawa CSU-W1 spinning disk unit, a 60× oil immersion objective lens (N.A. 1.49), a photometrics BSI sCMOS camera and Tokai Hit stage top incubator to maintain temperature, CO$_2$, and humidity. Lower magnification DSB imaging was performed on a BioTek Lionheart LX Automated Microscope, using ×40 magnification. For these DSB imaging experiments, neurons were cultured in Ibidi chamber slides (e.g., Ibidi #80826) with 300 μL of feeding volume.

## Neuronal purity staining and imaging

Neurons were fixed with 4% PFA, washed twice with PBS, then incubated with blocking buffer at RT for 30–60 min (5% normal goat serum and 0.1% Triton in PBS). After removing blocking buffer and washing twice with PBS, primary antibody solution was added (desired primary antibodies diluted appropriately in PBS with 3% normal goat serum) for a 1 h incubation at RT. After removing this solution and washing 3 times with PBS, secondary antibody solution was added (appropriate secondary antibodies diluted 1:500 in PBS with 3% normal goat serum, along with 1:1000 diluted DAPI), for a 1 h incubation at RT in the dark. Following 3 more PBS washes, PBS was added to the wells, and plates were stored foiled at 4 °C until ready to image.

Primary antibodies and their respective dilutions: rabbit anti-Ki67 (1:100, Abcam #ab16667), rat anti-NeuN (1:500, Abcam #ab279297), rabbit anti-TUBB3/Tuj1 (1:500, Sigma #T2200). Secondary antibodies, all used at 1:500 dilutions: goat anti-rabbit 488 (for DAPI-Ki67 combination, Invitrogen #A11008), goat anti-rat 488 (for DAPI-NeuN-TUBB3 combination, LifeTech #A11006), goat anti-rabbit 647 (for DAPI-NeuN-TUBB3 combination, Invitrogen #A21245). These imaging experiments were performed on a CellInsight CX7 microscope, with neurons cultured in PDL-coated black/clear 96-well plates (e.g., Corning # 354640). Images were analyzed by HCS Studio SpotDetector (Ki67 analysis) and CellProfiler (NeuN analysis).

## Flow cytometry

To dissociate neurons for flow cytometry, culture media was removed and then neurons were washed gently with PBS. Papain (reconstituted to 20 U/mL in PBS, Worthington #LK003178) was added and incubated for 10 min at 37 °C: 500 or 125 μL papain per well of a 6- or 24-well plate, respectively. Papain was then quenched with DMEM (Corning #10-013-CV) with 10% FBS (e.g., Avantor #1500-500 or Cytiva #SH30071.03) at 3–5× the papain volume, and pipetted around the edges to lift and collect the sheet of neurons. Neurons were then pelleted, resuspended in 100–500 μL of PBS per sample, and triturated to singularize. These samples were passed through strainer-capped FACS tubes (eg Stellar Sci #FSC-9005) and analyzed on an Attune NxT flow cytometer. Results were interpreted using FlowJo.

## Bulk RNAseq

For the initial RNAseq in iPSCs vs neurons, 3 replicate samples per condition were transduced in parallel using HIV VLPs. For the follow-up RNAseq with additional VLP controls in neurons, 4 replicate samples per condition were transduced in parallel using FMLV VLPs. Cells were transduced in 6-well plate format with 500,000 cells per well, using 25 μL of VLP in 2 mL of media per well. This dose corresponds to 1 μL of VLP per 20,000 cells, or 1.25 μL of VLP per 100 μL media.

Harvest timepoints for each cell type were selected based on their respective time courses of indel accumulation, per Fig. 2. The chosen timepoint for each cell type corresponds to when some, but not all, of the editing has occurred—so that DSB repair is actively ongoing at the time of harvest. This timepoint is 3 days post-transduction for neuron samples, and 1 day post-transduction for iPSC samples, which also avoids confounding effects from cell proliferation and/or dilution. Untransduced cells were harvested on the same day as the transduced cells.

RNA was extracted using the Zymo Quick-RNA™ Microprep Kit (cat. #R1050). For initial RNAseq: using 500 ng of total RNA per sample, we prepared the mRNA libraries using the QuantSeq 3′ mRNA-Sequencing Library FWD V1 Prep Kit (cat. #015.96). After cDNA synthesis, we used 17 PCR cycles to amplify the libraries. Following bead purification, mRNA concentrations were determined by Qubit and fragment lengths were quantified using High Sensitivity d5000 Reagents (cat. #5067-5593) on the Agilent TapeStation 4200. Normalized libraries were sequenced on the Illumina NovaSeqX 10B flow cell with parameters 101 × 12 × 24 (Read 1, Index 1, Index 2). Sequencing was performed at the UCSF Center for Advanced Technologies (CAT). For follow-up RNAseq: using 250 ng of total RNA per sample, we prepared the mRNA libraries using the QuantSeq 3′ mRNA-Seq V2 Library Prep Kit FWD with UDI 12 nt Set A1 (cat. #191.24). After cDNA synthesis, we used 16 PCR cycles to amplify the libraries. Following bead purification, mRNA concentrations were determined by Qubit and fragment lengths were quantified using Agilent 2100 Bioanalyzer (cat. #G2939B). After normalization, pooling, and dilution to a final loading concentration of 325 pM, samples were sequenced in-house on the Illumina NextSeq2000 using the P1 XLEAP-SBS™ flow cell and parameters 113x12x12 (Read 1, Index 1, Index 2).

Sequencing reads were trimmed using CutAdapt (DOI:10.14806/ej.17.1.200) and aligned using HISAT2 (DOI:10.1038/s41587-019-0201-4), and then a read count matrix was generated using featureCounts (DOI: 10.1093/bioinformatics/btt656). Differential expression analysis was performed on this count matrix using EdgeR (DOI:10.1093/bioinformatics/btp616). Functions within EdgeR used to statistically determine DEGs were: glmQLFit, glmQLFTest (with FDR for adjusted p-values), and decideTestsDGE. For venn diagrams and enrichment analysis, additional tools used were topTags with Benjamini–Hochberg correction for adjusted $p$-values, and WebGestalt (https://www.webgestalt.org/#).

## ChIP-qPCR

For each timepoint condition (untransduced, 2-day, 4-day, 8-day), 20 million neurons were grown across 2 10 cm dishes per condition (10 million neurons per plate). All 8 of these plates were cultured in parallel, during the same batch of differentiation. At Day 17, all transduced plates were transduced with 2 μL of B2Mg1 VLP (FMLV) per 100 μL media (total of 200 μL VLP per plate). Untransduced plates were harvested at Day 17. Remaining plates were harvested 2/4/8 days post-transduction as labeled.

At each harvest timepoint, two 10 cm dishes were fixed in parallel: one for each ChIP pulldown (Mre11 and γH2AX). Cells were fixed, pelleted, and snap frozen per ActiveMotif's ChIP fixation protocol (https://www.activemotif.com/documents/1848.pdf), then submitted

to ActiveMotif for ChIP-qPCR. ChIP-qPCR was performed using the antibodies and qPCR primers listed in Supplementary Data 8.

## Statistics and reproducibility

No statistical method was used to predetermine sample size. In amplicon-NGS, samples with insufficient aligned reads (e.g., < 60% aligned) were excluded. The experiments were not randomized. The investigators were not blinded during experiments and outcome assessment.

## Ethics statement

These studies involving human induced pluripotent stem cells were reviewed and approved by the UCSF Institutional Review Board. The donor from whom the WTC iPSC line was derived provided written informed consent for the generation and use of their iPSCs, which are commercially available through Coriell (GM25256). Experiments were performed in accordance with the relevant guidelines and regulations.

## Reporting summary

Further information on research design is available in the Nature Portfolio Reporting Summary linked to this article.

## Data availability

Source data for all figures and Supplementary Figs. are available via FigShare (https://figshare.com/articles/dataset/Source_Data_for_Characterizing_and_controlling_CRISPR_repair_outcomes_in_nondividing_human_cells_Ramadoss_et_al_Nature_Communications/30366298). Raw and processed RNA sequencing files have been deposited in the NCBI Gene Expression Omnibus (GEO) under accession code GSE272812 for the first set comparing iPSC and neuron samples, and under accession code GSE304183 for the second set evaluating additional control VLPs in neurons.

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

## Acknowledgements

We thank Zachary Nevin, Beeke Wienert, and all other members of the Conklin Lab for their helpful suggestions and feedback. We are also grateful for insightful advice from Jacob Corn, Alexis Komor, Jeffrey Hussmann, Ross Wilson, Hannah Karp, John Doench, Françoise Chanut, Netravathi Krishnappa, Michael Ward, Andy Qi, Hesong Han, Abdullah Syed, Bjoern Schwer, James Dahlman, Zoe Grant, David Toczyski, and Tippi Mackenzie. This work was enabled by the Gladstone Stem Cell, Flow Cytometry, Assay Development & Drug Discovery, and Bioinformatics Cores, as well as ActiveMotif, the IGI Next Generation Sequencing Core, and the UCSF Center for Advanced Technology. G.N.R. was supported by the NSF Graduate Research Fellowship, the UCSF Discovery Fellowship, and the Gladstone CIRM Scholars Program. J.R.H. was supported by NIH/NIGMS (K99GM143461-01A1) and the Jane Coffin Childs Memorial Fund for Medical Research. BLM was supported by NRSA (F32AG081085) and the L'Oréal USA For Women in Science Fellowship. M.S. was supported by the CIRM Postdoctoral Scholars Fellowship. C.F. was supported by a NIH/NIGMS Pathway to Independence Award (R00 GM118909) and a NIH/NIGMS Maximizing Investigators' Research Award (MIRA) for ESI (R35 GM143124). L.M.J. would like to acknowledge funding from R01 NS119678-01. B.R.S. was supported by NIH grants K08CA273529 and L30TR002983, the UCSF Living Therapeutics Initiative, the CIRM DISC-14907 grant and the CRISPR Cures for Cancer Initiative. The AN laboratory is supported by the Intramural Research Program of the NIH funded in part with Federal funds from the NCI under contract HHSN2612015000031. Research in the BA Laboratory was supported by the National Institutes of Health (R35GM138167).

NM would like to acknowledge funding from RO1MH125979-01, the HOPE NIH grant 1UM1AI164559, the Innovative Genomics Institute, the TED foundation, and the CRISPR-Cures grant. J.A.D. is an investigator of the Howard Hughes Medical Institute, and research in the JAD lab is supported by the Howard Hughes Medical Institute (HHMI), NIH/NIAID (U54AI170792, U19AI135990, UH3AI150552, U01AI142817), NIH/NINDS (U19NS132303), NIH/NHLBI (R21HL173710), NSF (2334028), DOE (DE-AC02-05CH11231, 2553571, B656358), Lawrence Livermore National Laboratory, Apple Tree Partners (24180), UCB-Hampton University Summer Program, Mr. Li Ka Shing, Koret-Berkeley-TAU, Emerson Collective and the Innovative Genomics Institute (IGI). M.K. was supported by Chan Zuckerberg Initiative grant 2022-316571. BRC was supported by the National Institutes of Health (R01-AG072052, R01-HL130533, R01-HL13535801, P01-HL146366), the California Institute for Regenerative Medicine (INFR6.2-15527), the Charcot-Marie-Tooth Association and by funding from Tenaya Therapeutics. B.R.C. acknowledges support through a gift from the Roddenberry Foundation and Pauline and Thomas Tusher.

## Author contributions

G.N.R. and B.R.C. conceived the study. B.R.C., M.K., and G.N.R. oversaw the overall project directions and planning. G.N.R., S.J.N., M.M.K., and K.G.C. designed and performed most experiments. M.S., B.L.M., and S.H.L. performed additional iPSC and neuron experiments. P.H.D. and M.P.M. performed cardiomyocyte experiments. L.A.W. and A.S.H. performed T cell experiments. J.R.H., C.F., B.S.P., and C.R.S.E. provided VLP formulations. R.S. and N.M. provided LNP formulations. J-.C.L. imaged DSB foci. R.S.B. and J.J. performed RNAseq library prep. H.L.W., L.M.J., B.R.S., A.N., B.A., and J.A.D. provided conceptual and critical guidance and helped shape the manuscript. GNR wrote the manuscript with input from all authors.

## Competing interests

M.K. is a co-scientific founder of Montara Therapeutics and serves on the Scientific Advisory Boards of Engine Biosciences, Casma Therapeutics, Cajal Neuroscience, Alector, and Montara Therapeutics, and is an advisor to Modulo Bio and Recursion Therapeutics. M.K. is an inventor on US Patent 11,254,933 related to CRISPRi and CRISPRa screening, and on a US Patent application on in vivo screening methods. JRH is a co-founder of Azalea Therapeutics. CF is a co-founder of Mirimus, Inc. B.A. is an advisory board member with options for Arbor Biotechnologies and Tessera Therapeutics. B.A. holds equity in Celsius Therapeutics. The Regents of the University of California have patents issued and pending for CRISPR technologies (on which J.A.D. is an inventor) and delivery technologies (on which J.A.D. and J.R.H. are co-inventors). J.A.D. is a cofounder of Azalea Therapeutics, Caribou Biosciences, Editas Medicine, Evercrisp, Scribe Therapeutics, Intellia Therapeutics, and Mammoth Biosciences. J.A.D. is a scientific advisory board member at Evercrisp, Caribou Biosciences, Intellia Therapeutics, Scribe Therapeutics, Mammoth Biosciences, The Column Group, and Inari. J.A.D. is Chief Science Advisor to Sixth Street, a Director at Johnson & Johnson, Altos and Tempus, and has a research project sponsored by Apple Tree Partners. The remaining authors declare no competing interests.
