## [Transparent Peer Review file · Nature Communications]

Characterizing and controlling CRISPR repair outcomes in nondividing human cells

Corresponding Author: Dr Bruce Conklin

Version 0:

Reviewer comments:

Reviewer #1

(Remarks to the Author)

Identifying the differences in DNA repair between different cell types—particularly in replicating and post-mitotic cells—is important for understanding the genome stability in studied cells and can guide future strategies aimed at therapeutic corrections of pathogenic alleles. In this manuscript, the authors focused on DNA repair of Cas9-induced DNA double-strand breaks (DSBs) in an isogenic pair of cells: proliferating induced pluripotent stem cells (iPSCs) and nondividing cortical-like excitatory neurons. The genome editing components were delivered using engineered virus-like particles (VLPs). The key finding is that in neurons, DNA DSBs are resolved much slowly compared to iPSCs, and, as a result, indels accumulate over prolonged time in the post-mitotic cells. Next, the authors compared the transcriptomic responses to VLP-mediated Cas9 delivery in iPSCs and neurons. Their analysis revealed unexpected differences between the two cell types and differentially expressed genes involved in DNA repair in transduced neurons. By small molecule- or siRNA-based inhibition of one of the most upregulated genes in transduced neurons, the authors demonstrated the possibility to improve both the efficiency and precision of targeted gene inactivation by Cas9. Although the key finding of the presented work is important and provides valuable insight for the field, a few major issues remain to be addressed and clarified prior to acceptance for publication.

A main weakness is that the study is entirely based on human iPS derived neurons. Primary neurons are not part of the study. This could readily be done with mice. Ideally include iPSC-derived neurons also. This way there is both stem cell derived and primary neurons in the same system and potential differences may be present.

Fig. 1c. There should be quantification of the foci, including over time. Similar for Fig. 2D,E. The measurement of indel products is not sufficient to evaluate repair of DSBs, there may be other products that are not seen. So this needs to be illuminated from multiple angles.

The other major weakness is that it is also unclear how authors distinguish between multiple cycles of cutting and accurate repair from inefficient and imprecise repair.

What do we know about the half life of Cas9 in these cells, and its activity over time?

In one of the models the authors draw the various outcomes, but they are not all evaluated experimentally.

The analysis of transcriptional targets and the follow up on RNR is innovative and interesting, skewing the repair towards deletions.

The authors seem to aim to improve DSB repair and indeed formation towards the end, with the promise to make this therapeutically feasible. Another view is that this may direct efforts towards base editing. This should be discussed.

Throughout their analyses, the authors are comparing the transcriptomic profiles of transduced and untransduced cells. Such a control is suboptimal, especially when potentially highly immunogenic VLPs are used as delivery vehicles (PMID: 33632278). There might be cell-to-cell differences in the level of stress experienced after the transduction, possibly simulating

the expression of certain genes, including those involved in DNA repair. Therefore, the proper controls should comprise of empty vectors as well as vectors carrying proteins other than Cas9 (PMID: 27595405).

Reference 3 in line 48 is cited for imprecise genome repair. It would be more appropriate to cite literature of author(s) who had shown this.

The lack of repair is a central element of this study, so it should be stated where this was observed previously.

Reviewer #2

(Remarks to the Author)

Ramadoss et al reported the changes over time in the percentage of indels caused by CRISPR/Cas9 in iPSCs and neurons. They also identified changes in gene expression in neurons upon CRISPR/Cas9 treatment. Based on these changes, they also identified compounds that facilitate the induction of indels. This is an important finding for DNA repair by CRISPR/Cas9 in neurons. However, there are several points that need to be addressed.

Major points:

1. The authors have determined the time and rate at which indel occurs and identified compounds that increase indel, but it is not clear whether this will lead to increased opportunities for precise therapeutic editing as the authors describe. The increase in efficacy should be demonstrated in vivo with the administration of the compound.
2. In addition to the above issues, if the indication for treatment is to be considered, the effects of base editing of gene mutations should be demonstrated, e.g., in base editing.
3. It is reasonable to define iPSC-derived neurons as representatives of non-dividing cells. To show that such characteristics of CRISPR/Cas9 treatment are remarkable in non-dividing cells, it is important to clarify whether similar phenomena can also be observed in other non-dividing cells. Human cardiomyocytes complete differentiation and proliferation in the early fetal period, and then continue to remain in the body as non-dividing cells, and they have similarities to neurons. Studies on iPSC-derived cardiomyocytes would further strengthen the authors' assertion.
4. If it is difficult to prepare mature iPSC-derived cardiomyocytes that are non-dividing, comparing various cell types using primary cultures in mice would provide very important evidence.
5. The controls should be set up correctly. Based on RNAseq results, the authors were identifying compounds that increase indels. However, there was no negative control since gene expression levels were compared with both VLP transduced neurons and un-transduced neurons. The authors' data may indicate stress changes due to VLP or Cas9 administration, rather than indel-related shared expression changes in indel formation of the three genes – B2M/NEFL/NT. VLPs containing scrambled gRNA and Cas9 should be used as negative control.
6. In Figure 1d)e), image data should be quantified and statistically processed.

Minor points:

1. It is difficult to understand what the illustrations in Figure 1g mean. An explanation that is easy for the reader to understand should be provided.
2. Please provide a more detailed explanation using diagrams etc. about the platform for detecting DSBs by labeling with γ H2AX and 53BP1, which is commonly used in Figure 1c) and subsequent sections.

Reviewer #3

(Remarks to the Author)

The study "Neuronal DNA repair reveals strategies to influence CRISPR editing outcomes" by Ramadoss and colleagues addresses a significant gap in understanding DNA repair mechanisms in postmitotic neurons. Understanding these mechanisms is crucial for improving CRISPR editing outcomes in these cells and opens opportunities for precise therapeutic editing. The authors used an isogenic iPSC line and neurons derived from that line to compare DNA repair mechanisms. They found that Cas9-induced indels accumulate slowly in neurons compared to isogenic iPSC. Cas9-treated neurons overexpress DNA repair genes, including factors associated with replication. The authors found that manipulating this response allowed for directing neuronal repair toward desired editing outcomes, albeit the outcome is gRNA dependent. The findings might have significant implications for therapeutic genome editing, particularly in treating neurodegenerative diseases. Although the study has great potential and the manuscript is well-composed, additional work is needed.

Major comments:

The authors studied how non-dividing cells, particularly iPSC-derived cortical-like excitatory neurons, repair Cas9-induced DNA damage. However, a mechanism for the DNA repair and slow indel accumulation in these neurons is lacking. The authors could employ, for example, recently published CRISPRi and CRISPRa screening assays to investigate the mechanisms behind DSB repair. Functional assays are also needed to confirm these findings in isogenic iPSC vs neurons.

The authors stated that Cas9-VLPs elicit a striking transcriptional response in neurons. However, they only used untransduced neurons as a control. The authors should include non-targeting VLP controls to rule out alternative causes for

the prolonged DSB.

Minor comments:

The study claims to use non-dividing neurons. While NeuN positivity and Ki67 negativity strongly suggest they are non-dividing, additional methods such as BrdU incorporation and cell cycle analysis are needed to confirm that the cells are indeed in the G0/G1 phase.

The authors stated that transient delivery to neurons is challenging. However, they did not provide experimental details on how specified VLPs were selected. The authors should provide a comparative analysis of the transduction efficiency between different conditions. Additionally, Figure S3f appears to be overexposed. The authors should present the phase contrast and mNeonGreen images separately.

I have not seen the off-target and genomic stability analysis. The authors should assess off-target effects and long-term genomic stability in edited postmitotic neurons. There is limited discussion of clinical implications and the potential risks of prolonged DSB. These need to be discussed.

Some figures lack the statistical analysis. Include quantitative data and statistical analysis across all assays.

Version 1:

Reviewer comments:

Reviewer #1

(Remarks to the Author)

This manuscript on the kinetics of DSB repair in iPS derived neurons is a valuable contribution to the field. The authors have done an outstanding job to address the comments. The manuscript is very carefully done, comprehensive and a pleasure to read. The authors show new data on DNA repair foci, Chip-Seq of DNA repair proteins, on the long-lived nature of Cas9 in nondividing cells, and the ability to affect editing in T cells and cardiomyocytes, the latter reproducing the slow kinetics of neurons. The responses to the criticism are effective and carefully prepared, and the manuscript is much improved. Congratulations to the excellent work. I have only minor comments.

Figure 2c does not go back down to baseline. Additional datapoints would be desirable.

Figure 2a if there is a way to modulate the X axis (or a blow-up), it would allow to appreciate the kinetics of repair in iPS kinetics a bit better. Its not instantaneous either.

Reviewer #2

(Remarks to the Author)

The authors responded this reviewer's comments appropriately.

Reviewer #3

(Remarks to the Author)

I have already reviewed this manuscript. The study, Characterizing and Controlling CRISPR Repair Outcomes in Nondividing Human Cells by Ramadoss and colleagues, shows that CRISPR-Cas9 repair outcomes in nondividing human neurons, cardiomyocytes, and T cells differ from those in dividing iPSCs, with neurons repairing DNA breaks more slowly and relying on non-canonical pathways. These pathways can be chemically or genetically manipulated to bias outcomes, which represents a significant advance, as most prior work has focused on dividing cells where standard repair mechanisms are dominant. The ability to achieve precise genetic editing in postmitotic cells has major therapeutic implications. The work is original, impactful, and suitable for publication. The authors have responded to all of my concerns, and I have no additional comments.

We thank the reviewers for their thoughtful comments, and we are delighted to submit this revised manuscript which has been much improved by their insights.

We share the reviewers' impression that this study opens the door to many exciting directions, including exploring cell-type-specific differences in other types of genome editing such as base or prime editing, translating these findings into in vivo therapies, and using CRISPRi screening to further dissect neuronal mechanisms of DNA repair. Indeed, we have already written proposals to pursue many of these directions in subsequent studies.

For the present study, guided by the reviewers' suggestions, we have performed and incorporated a large set of new experiments which strengthened both the mechanism and the clinical relevance. This includes: exploring whether or not our findings also apply to *primary* nondividing cells, assessing how much of the observed transcriptional response in neurons was due to DSBs vs Cas9 itself vs VLP delivery particles, and distinguishing between our two proposed models for why DSB repair was so prolonged in neurons. Our responses to each of the reviewers' suggestions are provided in-line below.

Reviewer 1:

1.1. "A main weakness is that the study is entirely based on human iPS derived neurons. Primary neurons are not part of the study. This could readily be done with mice. Ideally include iPSC-derived neurons also. This way there is both stem cell derived and primary neurons in the same system and potential differences may be present."

As Reviewer 1 and Reviewer 2 aptly point out, we are greatly interested in determining whether our DNA repair observations will also apply to other models beyond iPSC-derived neurons. However, mouse cells are a suboptimal model for this question. The DNA repair machinery of mice is known to be drastically different from that of humans. For example in the "Brain RNAseq" database (<https://brainrnaseq.org/>), expression of key DNA repair factors from our manuscript including RRM2, POLL, and XRCC5 differ by *orders of magnitude* in mouse neurons compared to human neurons. Mouse cells also have mouse DNA sequences that are different than the human DNA sequences, complicating comparisons between experiments. Testing our DNA repair findings in mouse neurons (either in vivo or ex vivo) is therefore less likely to reproduce, and likely *less* predictive of therapeutic relevance. Still, we wanted to address the reviewers' valid questions about whether our findings will be translationally relevant, and whether they will reproduce in other models besides iPSC-derived neurons – such as in primary cells. Therefore, rather than repeating our experiments in primary mouse cells, we instead repeated them in ***primary human cells***.

Specifically, we successfully repeated several key experiments from the paper in unactivated (nondividing) ***primary human T cells***. Using primary human T cells removes any concerns about immature iPSC-derived cells, while still being human cells instead of mouse cells. Also, in primary human T cells, Cas9 RNP can be delivered by electroporation, removing any delivery vehicles from the equation entirely. Ex vivo editing of primary human T cells is

currently being pursued clinically for next-generation CAR-T therapies – including inactivation of the B2M locus with our same B2Mg1 sgRNA. Therefore, editing in this primary human T cell model is **directly translatable** to the clinic. Intriguingly, recent reports suggested that such T cell editing should be performed in the unactivated (nondividing) state. Our new experiments in nondividing primary human T cells, inspired by our neuron experiments, explore the consequences of doing so.

The 3 key findings in primary human T cells are: 1. Nondividing vs dividing primary human T cells reproduced the differences in indel distribution (NHEJ-like vs MMEJ-like) observed in neurons vs iPSCs. 2. Nondividing vs dividing primary human T cells (with electroporation of Cas9 RNP) did NOT reproduce the dramatically slower indel accumulation seen in neurons vs iPSCs (with VLP delivery). This suggests that a component of the time course observed in neurons and cardiomyocytes may be dependent on cell type and/or delivery kinetics; we have incorporated new text stating this. 3. Nondividing primary human T cells reproduced the effects of RNR inhibition on Cas9 indels seen in neurons. RNR inhibition consistently shifted indels from insertions toward deletions, and boosted overall editing efficiency in an sgRNA-dependent manner. These results have been incorporated into Figure 3, Supplemental Figures S10, S11, and S22, and throughout the text of the manuscript.

Note: we also attempted several times to test our findings in primary human *neurons* harvested from surgical resections from epilepsy patients. We tested 4 transient delivery vehicles, but unfortunately none were able to reliably transfect neurons in this slice culture system. Therefore, our new primary human T cell data is the most “clinically relevant” model of nondividing cell editing that is currently available to us.

We hope the reviewers agree that this is even more translationally relevant than the initial suggestion, as *ex vivo* editing of primary human T cells is being pursued clinically as a therapy at this very moment – including editing of the exact locus we tested. Therefore, these new results strengthen our findings’ impact on therapeutic editing more broadly: not only future therapies in neurons, but also imminent therapies in T cells.

1.2. “Fig. 1c. There should be quantification of the foci, including over time. Similar for Fig. 2D,E”

Per the reviewer’s suggestion, we have incorporated quantification of the DSB staining in neurons into Figure 2: quantifying foci across several timepoints post-transduction, and multiple sgRNAs. The reason we do not quantify the snapshot in Figure 1c is because it would be misleading to quantify “percentage of neurons with a DSB” at a single snapshot in time, given what we know about the prolonged editing process in neurons. For example, if 40% of neurons have DSBs at day 3, and 40% of neurons have DSBs at day 7, with these endpoint assays it is impossible to determine whether this means 80% of neurons received a DSB, or only 40% (with the same neurons still showing unresolved DSBs at the later timepoint) – or somewhere in between. Since quantifying a single snapshot would be misleading, we keep 1c as simply a *qualitative* confirmation that we can induce DSBs in our postmitotic neurons – and we instead have now quantified the **full** time course of DSBs in Figure 2.

1.3. “The measurement of indel products is not sufficient to evaluate repair of DSBs, there may be other products that are not seen. So this needs to be illuminated from multiple angles.”

As the reviewer points out, measurement of indel products by sequencing alone is insufficient, as it cannot detect any DSB loci that have *not yet been sealed*. Therefore, a strength of our study is that we used multiple additional orthogonal assays capable of detecting these **unrepaired** products: ICC for DSB repair foci anywhere in the cell, and ChIP-qPCR for two different DSB repair proteins bound near the target site. These 3 distinct approaches (indels, ICC, and ChIP-qPCR) to quantifying DSB repair over time are highly complementary, as they each are capable of detecting products that the others would miss. For example, the unrepaired loci invisible to sequencing are exactly the loci detected by both ChIP-qPCR and ICC. Based on the reviewer’s comments, we have added new text emphasizing this complementarity.

Another potential product that would be missed by measurement of indel products alone is: DSB loci that have recently been repaired, but *without an indel*. To our knowledge, no assays have yet been developed to distinguish between “never-cut” loci and “cut-but-perfectly-repaired” loci. To address this, we incorporated an additional ChIP-qPCR amplicon which spans across the cut site. Detection of Mre11 or gH2AX bound to this site indicates that it was recently cut, but successful amplification across that cut site indicates that the locus is now intact (repaired).

This unique assay for detecting cut-then-repaired products, combined with sequencing which quantifies what fraction of “repaired” loci were repaired with vs without an indel, gives us a more complete picture of DSBs over time. We have restructured Supplemental Figure S13 and the corresponding text, to explain this assay.

1.4. “The other major weakness is that it is also unclear how authors distinguish between multiple cycles of cutting and accurate repair from inefficient and imprecise repair. What do we know about the half life of Cas9 in these cells, and its activity over time? In one of the models the authors draw the various outcomes, but they are not all evaluated experimentally.”

As the reviewer suggests, the most direct way of distinguishing between our two proposed models is to characterize the half-life and activity of Cas9 over time in our neurons. If Cas9 is absent or non-functional within a few days, yet DSBs remain unresolved for weeks, then this would more strongly support the model that neurons are “slow” to repair their DSBs. If Cas9 remains present and functional for a much longer time in neurons, then this would more strongly support the model that neurons experience multiple cycles of cutting and perfect repair until an indel finally occurs. Therefore, we utilized two new assays to address these questions, both incorporated in Supplemental Figure S14 as well as the results section text.

First, we used ELISA to quantify how much Cas9 protein was detectable in the lysates of Cas9-treated iPSCs and neurons, across several timepoints post-transduction. While VLP-delivered Cas9 was undetectable in iPSCs after 8 days, Cas9 remained detectable in neurons even after 30 days. This clearly demonstrates that the half-life of Cas9 protein is far longer than expected in neurons, likely due to the lack of dilution from cell division and the lack of nuclear envelope dissolution. However, this assay does not yet indicate how much of the remaining Cas9 protein is functional, as opposed to e.g. Cas9 in the process of being degraded.

Second, we therefore designed a new assay to quantify the longevity of “functional Cas9”. In this assay we first deliver Cas9-only VLPs with no sgRNA. We then deliver sgRNA-only LNPs at 1, 2, 4, or 8 days post VLP delivery; in each case we sequence the editing outcomes eight days post sgRNA-LNP delivery. If sgRNA delivered N days post-Cas9-delivery is still sufficient to yield detectable editing, this implies that “functional” Cas9 remains present in neurons for at least N days. By this assay, the level of “functional” Cas9 in our neurons drops off much more quickly than the level of “total” Cas9 from the ELISA assay. However, a significant amount of functional Cas9 still remains in our neurons for at least 8 days, as we observed 15% editing in neurons even when the sgRNA was only delivered 8 days after the Cas9.

So while we cannot *fully* rule out either model, these new experiments more strongly support our second model: that the prolonged DSB repair timeline in Cas9-treated neurons likely comes from neurons undergoing multiple cycles of cutting and repair until an indel arises. This is consistent with the increased longevity of Cas9 in nondividing cells, and the bias of neurons toward indel-free cNHEJ. We thank the reviewer for this valuable suggestion which greatly strengthened our study.

1.5. “The analysis of transcriptional targets and the follow up on RNR is innovative and interesting, skewing the repair towards deletions.”

Thank you!

1.6. “The authors seem to aim to improve DSB repair and indeed formation towards the end, with the promise to make this therapeutically feasible. Another view is that this may direct efforts towards base editing. This should be discussed.”

We agree with this possibility, and we are excited about the neuronal base editing results provided in Supplemental Figure S9c. In followup studies we are actively interested in determining the relative safety and efficacy of base editing in neurons, as ssDNA repair is known to be drastically different in neurons and may come with unique risks. We have incorporated the following statement into the discussion: “Future studies could use our tools to optimize the safety and efficacy of *DSB-independent* editing modalities in neurons as well.”

1.7. “Throughout their analyses, the authors are comparing the transcriptomic profiles of transduced and untransduced cells. Such a control is suboptimal, especially when potentially highly immunogenic VLPs are used as delivery vehicles (PMID: 33632278). There might be cell-to-cell differences in the level of stress experienced after the transduction, possibly simulating the expression of certain genes, including those involved in DNA repair. Therefore, the proper controls should comprise of empty vectors as well as vectors carrying proteins other than Cas9 (PMID: 27595405).”

We thank the reviewers for this important suggestion. To better distinguish how much of the observed transcriptomic profile was in response to DSBs, vs Cas9 protein, vs VLP delivery vehicles themselves, we have now repeated the RNAseq in neurons with key additional control groups. We show these results in Figure 3 and Supplemental Figure S19, using VLPs that deliver: Cas9-B2Mg1, Cas9-NTg1 (non-targeting), **dCas9**-B2Mg1, or GFP. We also added new text describing these results and the main takeaways: a component of this response is indeed

triggered by VLPs themselves, but the presence of Cas9 DSBs significantly amplifies the transcriptional response.

For example, RRM2 is significantly more upregulated in neurons treated with a cutting Cas9-VLP, compared to either dCas9-VLP or non-targeting Cas9-VLP. We appreciate the reviewers recommending these additional controls, which have improved our understanding of the neuronal response.

1.8. “Reference 3 in line 48 is cited for imprecise genome repair. It would be more appropriate to cite literature of author(s) who had shown this. The lack of repair is a central element of this study, so it should be stated where this was observed previously.”

We did not intend for this sentence to reference the possibility of “lack of repair”, but rather to point out that DSBs can be repaired in unproductive or even harmful ways. Reference 3 nicely summarizes dozens of studies that explore these different types of undesired repair outcomes for various editing modalities (including Cas9, base/prime editing, and more), and specifically discusses some of the harmful consequences of inaccurate DSB repair.

Reviewer 2:

2.1. “The authors have determined the time and rate at which indel occurs and identified compounds that increase indel, but it is not clear whether this will lead to increased opportunities for precise therapeutic editing as the authors describe. The increase in efficacy should be demonstrated in vivo with the administration of the compound.”

In response 1.1 above, we detail the new experiments we performed to address this excellent question raised by both Reviewer 1 and Reviewer 2.

2.2. “In addition to the above issues, if the indication for treatment is to be considered, the effects of base editing of gene mutations should be demonstrated, e.g., in base editing.”

Reviewer 2 shares Reviewer 1’s interest in how our findings may impact base editing in neurons, which we discuss in response 1.6 above.

2.3. “It is reasonable to define iPSC-derived neurons as representatives of non-dividing cells. To show that such characteristics of CRISPR/Cas9 treatment are remarkable in non-dividing cells, it is important to clarify whether similar phenomena can also be observed in other non-dividing cells. Human cardiomyocytes complete differentiation and proliferation in the early fetal period, and then continue to remain in the body as non-dividing cells, and they have similarities to neurons. Studies on iPSC-derived cardiomyocytes would further strengthen the authors’ assertion.”

We thank the reviewer for this suggestion, and we have therefore reproduced multiple key experiments in postmitotic human iPSC-derived cardiomyocytes. In Supplemental Figure S9a we show that cardiomyocytes share neurons’ prolonged accumulation of indels. In Supplemental Figure S24 we show that the DNA repair perturbations we identified in neurons have the same effects on genome editing in cardiomyocytes. As the reviewer suggests, we feel these results greatly strengthen our findings and broaden the impact of our study.

2.4. “If it is difficult to prepare mature iPSC-derived cardiomyocytes that are non-dividing, comparing various cell types using primary cultures in mice would provide very important evidence.”

We successfully tested these findings in postmitotic iPSC-derived cardiomyocytes per the reviewer’s initial suggestion, as described in 2.3. Therefore, we have now explored DSB repair in nondividing human iPSC-derived neurons, nondividing human iPSC-derived cardiomyocytes, and nondividing human primary T cells.

2.5. “The controls should be set up correctly. Based on RNAseq results, the authors were identifying compounds that increase indels. However, there was no negative control since gene expression levels were compared with both VLP transduced neurons and un-transduced neurons. The authors' data may indicate stress changes due to VLP or Cas9 administration, rather than indel-related shared expression changes in indel formation of the three genes □ B2M/NEFL/NT. VLPs containing scrambled gRNA and Cas9 should be used as negative control.”

As detailed in response 1.7 above, we have now incorporated several new RNAseq controls to better distinguish how much of the transcriptional response was related to DSBs vs administration of Cas9 or VLPs. Note: NTg1 is a non-targeting sgRNA (similar to scrambled), as opposed to a gene.

2.6. “In Figure 1d)e), image data should be quantified and statistically processed.”

In response 1.2 above, we describe in more detail how we have now quantified the images of DSB foci over time.

2.7. “It is difficult to understand what the illustrations in Figure 1g mean. An explanation that is easy for the reader to understand should be provided.”

Per the reviewer’s suggestion (assuming this refers to Figure 2g), we have added a more thorough explanation of the two models: in the text of the results section, and in the figure caption. We have also slightly modified the illustration itself.

2.8. “Please provide a more detailed explanation using diagrams etc. about the platform for detecting DSBs by labeling with γ H2AX and 53BP1, which is commonly used in Figure 1c) and subsequent sections.”

We apologize for any lack of clarity. The detection of γ H2AX and 53BP1 was performed simply by antibody staining (immunocytochemistry) and imaging, described in the methods section under “DSB marker staining and imaging”.

Reviewer 3:

3.1. “The authors studied how non-dividing cells, particularly iPSC-derived cortical-like excitatory neurons, repair Cas9-induced DNA damage. However, a mechanism for the

DNA repair and slow indel accumulation in these neurons is lacking. The authors could employ, for example, recently published CRISPRi and CRISPRa screening assays to investigate the mechanisms behind DSB repair. Functional assays are also needed to confirm these findings in isogenic iPSC vs neurons.”

We share the reviewer’s enthusiasm for more deeply investigating neuronal mechanisms for repairing Cas9-induced damage. In response 1.4 above, we describe the two new assays we have now incorporated to clarify the mechanism of slow indel accumulation in neurons. In the present study, we have also used arrayed screening to begin probing the mechanisms of neuronal DSB repair, currently performing genetic knockdowns via siRNA inside all-in-one LNPs (rather than CRISPRi). These results are shown in Figure 4 and Supplemental Figure S24.

In ongoing followup studies, we are using high-throughput CRISPRi screening to further investigate mechanisms of neuronal DSB repair – as hinted at in the discussion section. These screening studies comprise entire projects of their own; we are excited about these impactful directions that have already been enabled by the foundational findings of this first study.

3.2. “The authors stated that Cas9-VLPs elicit a striking transcriptional response in neurons. However, they only used untransduced neurons as a control. The authors should include non-targeting VLP controls to rule out alternative causes for the prolonged DSB.”

We agree with this assessment, and we have therefore incorporated several new controls to decipher how much of this neuronal response was due to DSBs specifically, vs alternative causes. These new controls (and the results) are described in full detail above, in response 1.7.

3.3. “The study claims to use non-dividing neurons. While NeuN positivity and Ki67 negativity strongly suggest they are non-dividing, additional methods such as BrdU incorporation and cell cycle analysis are needed to confirm that the cells are indeed in the G0/G1 phase.”

Per the reviewer’s request, we have now included BrdU staining as a third marker confirming our cells’ identity as nondividing neurons. Supplemental Figure S2i-j now shows that BrdU incorporation was detected in iPSCs but not in neurons. (Also note: Supplemental Table 1 contains the RNAseq expression results for every gene in every sample, including untreated control neurons. These results additionally confirm the lack of expression of many key cell cycle markers in our neurons, at population-level.)

3.4. “The authors stated that transient delivery to neurons is challenging. However, they did not provide experimental details on how specified VLPs were selected. The authors should provide a comparative analysis of the transduction efficiency between different conditions.”

We thank the reviewer for pointing this out. We have now changed Supplemental Figures S3 and S4 to include more thorough details of how we selected these VLPs. Especially in Supplemental Figure S4 (and the corresponding text), we now provide the relative transduction efficiency of multiple VLP types, including the effect of varying both the VLP

pseudotype and NLS. These results demonstrate how each of these different parameters can impact the efficiency of VLP delivery in neurons.

Based on this excellent suggestion, we also included similar data to show how our specified **LNPs** were selected as well. These changes are reflected in Figure 4 and Supplemental Figure S23, as well as the corresponding text. These additional details may be informative for any readers interested in exploring VLP or LNP delivery in neurons or other cell types.

3.5. “Additionally, Figure S3f appears to be overexposed. The authors should present the phase contrast and mNeonGreen images separately.”

We agree, and we have removed this figure panel, since it does not provide additional information compared to the quantitative flow cytometry that follows it.

3.6. “I have not seen the off-target and genomic stability analysis. The authors should assess off-target effects and long-term genomic stability in edited postmitotic neurons. There is limited discussion of clinical implications and the potential risks of prolonged DSB. These need to be discussed.”

We agree that our findings indeed may have clinical implications, especially regarding the risks of prolonged DSBs affecting genomic off-targets and genomic stability. Therefore, we have added new text throughout the results and discussion sections to explicitly discuss these implications.

Note: since none of the sgRNAs used in this study are therapeutic targets in neurons, measuring the off-targets of these particular sgRNAs in neurons would be irrelevant. (We chose biallelic-targeting sgRNAs against neuron disease genes, whereas the actual therapies would need to be allele-specific sgRNAs selectively inactivating the dominant mutant allele.) However, we absolutely agree that any sgRNAs being pursued for a therapy must first be thoroughly assessed for off-target effects and genomic stability. Furthermore, our findings suggest that this assessment should be performed in the correct cell type model (not dividing cell lines), and at later time points than previously anticipated. We have adjusted the text to mention these takeaways more clearly.

3.7. “Some figures lack the statistical analysis. Include quantitative data and statistical analysis across all assays.”

We have now updated figure panels to include quantification and statistics where appropriate, as well as showing individual replicates for most bar graphs. For overlaid indel histograms throughout the manuscript, for which statistical comparisons are difficult to display, we have now incorporated corresponding violin plots for each panel, which quantify the ratio of insertions to deletions. These violin plots include statistical analysis between the groups in question.